# Investigation into owner-reported differences between dogs born in versus imported into Canada

**Kai Alain von Rentzell**[1]*, **Karen van Haaften**[2], **Amy Morris**[2¤], **Alexandra Protopopova**[1]

**1** The Animal Welfare Program, Faculty of Land and Food Systems, University of British Columbia, Vancouver, Canada, **2** British Columbia Society for the Prevention of Cruelty to Animals, Vancouver, Canada

¤ Current address: Vancouver Humane Society, Vancouver, Canada
* kaiv@student.ubc.ca

## Abstract

Over 1 million dogs are imported into the United States and roughly 340,000 dogs into the United Kingdom yearly. Although the official number of dogs arriving to Canada is currently unknown, local animal professionals estimate that thousands of dogs are imported into Canada each year. Dog importation may be increasing globally while regulation and surveillance are still limited, resulting in concerns for the health and welfare of imported dogs. To date, few studies have investigated how the source location of dogs influences the owner-dog relationship. The current report presents two independent studies that were conducted to assess whether owners of imported dogs reported a poorer owner-dog relationships compared to owners of Canadian-born dogs. In both studies, an online survey was distributed to dog owners (Study 1: n = 803; Study 2: n = 878) in British Columbia, Canada, containing questions on various aspects of the owner-dog relationship. The first study included questions from the Lexington Attachment to Pets Scale, Canine Behavioral Assessment and Research Questionnaire, Human-Animal Bond questionnaire, Monash Dog Owner Relationship Scale, and constructed questions about training methods, expectations, and health. The second study was comprised of original questions assessing difficult behaviour, training practices, health, attachment, and perceived level of burden of owning a dog. Both studies found no evidence of a poorer owner-dog relationship in non-Canadian-sourced dogs. In fact, owners of Canadian-sourced dogs used harsh training methods more frequently and had higher expectations for their dog. While no signs of poorer owner-dog relationship in non-Canadian-sourced dogs were found, future research should continue the investigation of age, health, and backgrounds of incoming dogs.

## Introduction

Each year, an unknown number of dogs are transported across significant distances and into new countries. It is estimated that roughly 1.06 million dogs are imported into the United States each year, of which 700,000 arrive by air and 360,000 arrive on land [1]. Additionally,

**Data Availability Statement:** All relevant data are within the manuscript and its Supporting information files.

**Funding:** This study was funded, in part, by the Natural Sciences and Engineering Research

Council of Canada and the British Columbia Society for the Prevention of Cruelty to Animals (#Industrial Research Chair 554745-19). https:// www.nserc-crsng.gc.ca/index_eng.asp https:// spca.bc.ca/ The authors Karen van Haaften and Amy Morris were employees of the BC SPCA at the time of the study. Both of the authors played a role in study design, data collection, and preparation of the manuscript.

**Competing interests:** The authors have declared that no competing interests exist.

roughly 44,000 dogs were imported into the United Kingdom for commercial purposes in 2019 [2] with an additional 300,000 dogs being imported non-commercially each year [3, 4]. Moreover, data from currently available reports suggest there to be a global increase in the number of dogs that are being transported between countries each year [2, 5]. One reason for the increase in cross-country movement is an increase in the 'resale' of dogs, including adoption, sales, and other transfers of ownership [5].

Dog imports to Canada are governed by two agencies: The Canadian Food Inspection Agency (CFIA), which is responsible for establishing regulations for all import animals [6], and the Canadian Border Services Agency, which is responsible for inspecting the animals and enforcing these standards [7]. Consequently, official statistics on dog importation are currently unavailable for Canada since there is no governmental agency responsible for tracking the number of arriving dogs. The Canadian National Canine Importation Working Group estimated that at least 6,189, but likely many more, dogs were imported into Canada from 29 countries through rescue organizations in 2013 [8]. The Working Group was concerned about the limited control for good health of imported dogs. Reports of dogs arriving sick or developing illness following arrival has resulted in heightened concerns regarding the risks to animal health, public health, and buyer/adopter satisfaction and wellbeing [9–13]. This concern was further exacerbated with the spike in demand for puppies during the COVID-19 pandemic, causing owners to make online purchases, some which may have been from uncertified sellers in foreign countries [14]. In response to zoonotic and welfare concerns, the CFIA updated import requirements for commercial dogs below 8 months of age entering Canada in May 2021. Nonetheless, there remains concerns of zoonotic risks from dog importation, since even one case can result in a zoonotic outbreak that may have serious consequences for animal and human health.

A frequently discussed risk of dog importation within the veterinary literature is the potential for the introduction of novel zoonotic diseases to communities [3, 15–17]. Canine rabies has a long incubation period, which complicates detection and poses risks for dogs as well as public health [15]. While rabies has been successfully eliminated from North America and most of Europe, rabies is still endemic in many developing countries [18, 19]. Importation of dogs from rabies-endemic countries presents a threat of re-introduction of rabies if preventative measures have not been taken. Dogs originating from Eastern European countries present a particular concern as this region is a major supplier of puppies for neighbouring countries [20]. Furthermore, dogs arriving from these regions have the highest frequency of inaccurate vaccination certificates [14], potentially suggesting occurrences of illegal activities [21]. Other zoonotic infectious disease concerns associated with the global movement of dogs include *Leishmaniasis spp.* [22, 23], *Echinococcus spp.* [24, 25], and exotic ticks or vector-borne diseases [15, 26], among many others [3, 27].

There is also heightening concerns that imported puppies, particularly those that have been purchased from abroad may be coming from countries where animal husbandry practices are largely unregulated [14], or come from large-scale intensive breeding facilities which may be engaging in poor animal husbandry practices such as breeding of dogs susceptible to the development of health and behavioural complications [20, 28]. Potential reasons for such poor outcomes may be that large-scale commercial dog breeders select dams and sires based on physical traits while temperament and health may be a secondary consideration [28]. Additionally, these dogs may be subjected to a number of stressors, such as early weaning, transportation, handling by numerous unfamiliar individuals, and relocation, during crucial early developmental stages of their lives, potentially increasing their likelihood of developing difficult-to-manage behaviours in adulthood [29–31]. Previous research has found that dogs obtained from large-scale commercial breeders tended to be at a greater risk for the development of aggression or anxiety-related problems [29, 30].

Despite widespread concerns about the importation of dogs, research on the experiences of the owners of these dogs is limited. While "problematic" behaviours in dogs are frequently reported as a reason for relinquishment [32, 33], there is a growing body of research investigating the owner-dog relationship to identify the possible roots of incompatibility. Recent findings suggest that owner perceptions and attitudes may have a larger influence on the owner-dog relationship than dog characteristics [32, 34–36]. Numerous questionnaires and scales have been developed to evaluate various aspects of the owner-dog relationship, such as the Monash Dog Owner Relationship Scale (MDORS) assessing owner-perceived cost and benefits of the relationship [37], Lexington Attachment to Pets Scale (LAPS) evaluating owner's level of attachment to their pet [38], and the Human-Animal Bond (HAB) assessing numerous facets including the owner's level of satisfaction with their pet [39], among many others. The current study incorporated questions from previously validated questionnaires to assess the owner-dog relationship.

Two separate studies were conducted to investigate the effect of source on the human-dog relationship. The aim of Study 1 was to investigate owner-reported differences in the owner-dog relationship between dogs sourced from Canada compared to dogs sourced from outside of Canada. The study examined various aspects of the owner-dog relationship to investigate whether non-Canadian dogs are at a greater risk for a compromised owner-dog relationship, and subsequently dog welfare, compared to domestic counterparts. The aim of Study 2 was to extend the findings of the first study. As such, Study 2 examined the same owner-dog relationship dimensions on a simplified scale through a separate set of respondents. Additional dog characteristics that were included were: dog size, dog breed, and dog age upon acquisition.

## Study 1

### Methods

**Survey design.** The survey collected owner demographic information (age, level of education, gender, number of adults in the household, number of children in the household) and basic information of the dog (dog age, country of origin, source method, puppy background), which was used to examine the effect of owner and dog factors on the owner-dog relationship. The number of adults in the household, and number of children in the household, and dog age were collected as numerical values. The owner's age, owner's level of education, owner gender, dog's country of origin, dog source method, and puppy background were collected as categorical variables. Dog source method contained seven options from which the participants selected the category that best described the method that was used to acquire their dog (1: Shelter/rescue/vet clinic; 2: Purebred breeder; 3: Friend/relative/neighbour; 4: Through online/print/in person advertisement from stranger; 5: Pet store; 6: Offspring of another dog in my household; 7: Found as a stray). Puppy background contained nine options from which the participant selected the category that best described their dog's background as a puppy (1: Born to dog from Canadian breeder (intentional litter); 2: Don't know; 3: Friend/neighbor/relative's dog had puppies; 4: Born in animal shelter/rescue/vet clinic; 5: Born to stranger's dog (accidental litter); 6: Born to dog living on streets/found/stray; 7: Born to dog from international breeder (intentional litter); 8: Born to dog in my household; 9: Pet store). International rescue was a variable created using the owner responses. International rescue was coded "Yes" if non-Canadian dogs were acquired through either a "Shelter/rescue/vet clinic", "Found as a stray", or "Through online/print/in person advertisement from stranger" as indicated from their dog source method. In all other instances, this international rescue was coded as "No". These measurements were included in the analysis as independent variables.

To illustrate the multifaceted nature of the owner-dog relationship, the current study assessed seven owner-dog relationship (ODR) dimensions: (1) perceived behavioural problems, (2) perceived cost of care, (3) satisfaction with their dog, (4) training methods, (5) perceived health of their dog, (6) expectation of their dog, and (7) attachment to their dog. The owner-dog relationship survey questions were constructed by integrating questions from previously validated questionnaires, including the Canine Behavioral Research Questionnaire (C-BARQ) [40], Monash Dog Owner Relationship Scale (MDORS) [37], Lexington Attachment to Pet Scale (LAPS) [38], and the Human-Animal Bond questionnaire (HAB) [39], along with original questions that were created for the purpose of the study. These owner attitudes were used for the determination of overall strength of owner-dog relationship, which was subsequently used to determine whether imported dogs are at a greater risk of poor outcomes compared to domestic dogs.

The types of resources that were accessed by the owner were measured on a 5-point Likert scale, which was later converted into binary data ("yes", "no"). These questions are grouped as Dog care questions for the purpose of clarity. Dog care questions composed of a total of four questions: whether the owner took the dog for a veterinary visit within the last year (Vet visit), and the modes of assistance accessed when training their dog (a professional trainer, the internet, and previous experience). The full survey as well as the summary of survey components can be found in the (S1 Appendix and S1 File).

**Recruitment and respondents.** The online survey was launched in December 2019. The survey was distributed to a paid panel of online respondents currently residing in British Columbia (BC) using the surveying tool SurveyGizmo, now titled Alchemer (Louisville CO, USA: Widgix, LLC). The age, education, and gender distributions of the respondents were set to match that of the population of BC. Accordingly, a minimum sample size of 803 responses was set to meet the consensus quota for age, gender, and education distribution of BC. Inclusion criteria were currently being a resident of BC, an owner of at least one dog, and owning the dog for at least six months. All responses were received in January 2020. Each participant was reimbursed for completing the survey according to their panel payment schedule. The survey was approved by the UBC Behavioural Research Ethics Board (H19-03874).

**Statistical preparation.** All data were handled and analyzed using R version 4.0.1. The original dataset and R code used in the analysis can be found in the (S1 Dataset and S1 Code). Owner and dog parameters were used as independent variables in the statistical analyses. These responses were entered as categorical or numerical variables according to their data type into multivariate multiple regression and logistic regression models (S1 File).

Exploratory factor analysis (EFA) was conducted separately on each ODR dimension using its raw numerical Likert responses. The number of factors for each dimension was determined through the visual interpretation of the scree plot, evaluation of eigenvalues, and to minimize cross-loading on multiple factors. A rule-of-thumb factor loading cutoff of >.30 was selected in reference to previous studies using EFA to interpret C-BARQ responses [31]. Two items from the C-BARQ (Cag9. & Cag10.) and one item from the MDORS (M9.) were removed from the analysis due to 244 missing values and weak factor loadings respectively. EFAs of the ODR dimensions resulted in a total of 11 factors (C-BARQ: 2, MDORS: 2, HAB: 2, Training methods: 2, Perceived health issues: 1, Expectation: 1, LAPS: 1). Every factor was given a name descriptive of the items it contained to ease interpretation (Table 1). EFA Factor scores were generated using the weighted average method (i.e., regression scores), using varimax rotation.

All factors extracted from the EFA were also tested for their internal consistency using Cronbach's coefficient alpha. All factors except Training methods factor 2 ("Gentle training", alpha: .55) and Expectation factor ("Expectation", alpha: .61) had good internal consistency of alpha >.70. For practical purposes, we retained all variables for analyses, given that a

**Table 1. Owner-dog relationship dimensions and extracted factors.**

| ODR dimensions | Definition | EFA factors and interpretation |
|---|---|---|
| Perceived behaviour | Owner-perceived behavioural problems displayed by their dog. Survey questions were adopted from C-BARQ. | Factor 1: *"Difficult behaviour"* The level of owner-perceived difficult behaviours displayed by their dog. |
| | | Factor 2: *"Excitability"* The level of owner-perceived excitability of their dog. |
| Perceived cost | Owner-perceived cost of care of their dog. Survey questions were developed from MDORS. | Factor 1: *"Struggle"* The owner-reported level of struggle associated with dog ownership. |
| | | Factor 2: *"Burden"* The owner-reported level of burden experienced from dog ownership. |
| Satisfaction | Owner-reported level of satisfaction with their dog. Survey questions were developed from HAB Questionnaire. | Factor 1: *"Regret"* The owner-reported level of regret experienced from dog ownership. |
| | | Factor 2: *"Satisfaction"* The owner-reported level of satisfaction experienced from dog ownership. |
| Training methods | The types of training methods utilized by owners during training sessions with their dog. Original survey questions were developed for the purpose of this study. | Factor 1: *"Harsh training"* The owner-reported use of primarily punishment-based training methods during dog training sessions. |
| | | Factor 2: *"Gentle training"* The owner-reported use of primarily reward-based training methods during dog training sessions. |
| Perceived health | Owner-perceived health of their dog. Original survey questions were developed for the purpose of this study. | Factor 1: *"Perceived health"* The owner-perceived health condition of their dog. |
| Expectation | Owner expectations of their dog during acquisition. Original survey questions were developed for the purpose of this study. | Factor 1: *"Expectation"* The owner-reported level of expectations of their dog during acquisition. |
| Attachment | Owner-reported level of attachment to their dog. Survey questions were developed from LAPS. | Factor 1: *"Attachment"* The owner-reported level of attachment to their dog. |

Owner-Dog Relationship (ODR) dimensions, their definitions, and respective factors with their definitions extracted from each ODR dimension.

minimum of four variables was needed to conduct an EFA. Thus, caution must be taken when interpreting the factors with these low alpha scores. Low internal consistency among these two factors may be a result of limited items contained within each factor [41]. The results from the EFA and its Cronbach's alpha values can be found in the (S1 File).

**Statistical analysis.** The statistical analyses were exploratory in nature. Multivariate multiple regression models were used to test the effect of owner and dog parameters on the ODR dimension factor scores. Accordingly, owner and dog parameters were entered as predictor variables into the regression models. ODR dimensions that yielded only one factor (Perceived health, Expectation, and LAPS) were initially entered into an analysis of variance (ANOVA) to examine the effect of owner and dog parameters. Predictor variables that yielded statistically significant effects were then closely examined using a linear regression model. ODR dimensions that yielded two factors (C-BARQ, MDORS, HAB, and Training methods) were jointly entered into a multivariate analysis of variance (MANOVA) to examine the effect of owner

and dog parameters on both factors collectively. Predictor variables that yielded statistically significant effects were then closely examined for their effect on each factor using multivariate multiple regression models.

A logistic regression model was used to test the effect of owner and dog parameters on the Dog care question outcomes. Four separate logistic regression models were used to test each Dog care question. Statistically significant interactions are presented using odds ratio plots.

A reference variable was specified for each categorical (non-numerical) owner and dog parameters when they were entered into regression models. Specifically, the reference variables for the categorical owner and dog parameters were: owner gender (Female, n = 415), owner education (Primary/Secondary, n = 52), dog origin (non-Canadian, n = 58), dog source (Pure-bred breeder, n = 206), puppy background (Canadian dog breeder, n = 324).

## Results

**Descriptive statistics.** A total of 2,534 respondents interacted with the survey, of which 1,620 (63.9%) were disqualified for not meeting the inclusion criteria. A further 111 (4.4%) responses were dismissed from the data analysis due to partial completion of the survey, resulting in a total of 803 (31.7%) responses included in the study. Approximately half 51.7% (n = 415) were female, 47.9% (n = 386) were male, and <1% (n = 2) identified with an unspecified category. The largest proportion of the respondents fell into the age group "36 to 55" (40.5%, n = 325), followed by "56 to 79" (26.2%, n = 210). The majority of respondents answered "high school/trade school" (44.7%, n = 359) and "university degree" (37.9%, n = 304) for their highest level of education received. Most of the respondents were owners of Canadian dogs (92.7%, n = 745). Interestingly, 13 owners of Canadian dogs responded their dog to have been born to a dog from an international dog breeder. If not a mistake, these responses may be from owners of first-generation Canadian dogs, whose dam or sire was originally from a non-Canadian country. These responses were included in the analysis as Canadian dogs. Canadian dogs were most commonly obtained through shelters, rescue organizations, or veterinary clinics and born to a dog from a Canadian breeder (Tables 2 and 3). Of the respondents that were owners of foreign dogs (7.2%, n = 58), a majority of the dogs came from the United States (n = 35), followed by South Korea (n = 6), Mexico (n = 4), among other countries (S1 File). The majority of non-Canadian dogs were acquired through shelters, rescue organizations, or veterinary clinics (Table 2).

**Owner-Dog relationship dimensions as a function of owner and dog parameters.** Increase in owner age predicted a decrease in the "Difficult behaviour" score (Table 4). Male owners were more likely than female owners to predict higher "Difficult behaviour" scores. Owners with dogs that were acquired from a friend, relative, or a neighbour had higher "Difficult behaviour" scores compared to dogs that were acquired from a purebred breeder. Dogs that came from a shelter, rescue, or a veterinary clinic as a puppy, dogs that were born to a stranger, and dogs that were born to with unknown backgrounds scored lower for the "Difficult behaviour" scores compared to dogs that were born to a dog from a Canadian breeder.

Increase in dog age predicted a decrease in the "Excitability" factor score (Table 5). Dogs that were born to a dog from an international breeder scored lower on the "Excitability" factor compared to those that were born to a dog from a Canadian breeder.

Owners of Canadian dogs reported higher "Struggle" factor scores than owners of non-Canadian dogs (Table 6). Increase in owner age predicted a decrease in the "Struggle" score. Owners with high school or trade school, university, or postgraduate education compared to owners who responded "Primary school/Secondary school" for their highest level of education received, and male owners compared to female owners scored higher "Struggle" scores.

**Table 2. Dog source methods.**

| Source | Count | Percentage |
|---|---|---|
| **Canadian dogs** | **745** | **92.78** |
| Shelter/rescue/vet clinic | 202 | 27.11 |
| Purebred breeder | 195 | 26.18 |
| Friend/relative/neighbour | 161 | 21.61 |
| Through online/print/in-person advertisement from stranger | 121 | 16.24 |
| Pet store | 38 | 5.10 |
| Offspring of another dog in my household | 17 | 2.28 |
| Found as stray | 11 | 1.48 |
| **Non-Canadian dogs** | **58** | **7.22** |
| Shelter/rescue/vet clinic | 29 | 50.0 |
| Purebred breeder | 11 | 18.97 |
| Friend/relative/neighbour | 9 | 15.52 |
| Through online/print/in-person advertisement from stranger | 2 | 3.45 |
| Pet store | 3 | 5.17 |
| Offspring of another dog in my household | 3 | 5.17 |
| Found as stray | 1 | 1.72 |
| **Total** | **803** | **100** |

Number (count) and percentage of respondents stating the method utilized by owners when obtaining their dog.

**Table 3. Dog source methods as a puppy.**

| Puppy background | Count | Percentage |
|---|---|---|
| **Canadian dogs** | **745** | **92.78** |
| Born to dog from Canadian breeder (intentional litter) | 324 | 43.49 |
| Don't know | 112 | 15.03 |
| Friend/neighbour/relative's dog had puppies | 118 | 15.84 |
| Born in animal shelter/rescue/vet clinic | 61 | 8.19 |
| Born to stranger's dog (accidental litter) | 54 | 7.25 |
| Born to dog living on streets/found/stray | 23 | 3.09 |
| Born to dog from international breeder (intentional litter) | 13 | 1.74 |
| Born to dog in my household | 21 | 2.82 |
| Pet store | 19 | 2.55 |
| **Non-Canadian dogs** | **58** | **7.22** |
| Born to dog from Canadian breeder (intentional litter) | 0 | 0 |
| Don't know | 15 | 25.86 |
| Friend/neighbour/relative's dog had puppies | 7 | 12.07 |
| Born in animal shelter/rescue/vet clinic | 1 | 1.72 |
| Born to stranger's dog (accidental litter) | 3 | 5.17 |
| Born to dog living on streets/found/stray | 12 | 20.69 |
| Born to dog from international breeder (intentional litter) | 16 | 27.59 |
| Born to dog in my household | 2 | 3.45 |
| Pet store | 2 | 3.45 |
| **Total** | **803** | **100** |

Number (count) and percentage of respondents stating from where their dog came as a puppy.

**Table 4. Statistical significance of owner and dog parameters on CBARQ Factor 1 "Difficult behaviour".**

| Parameter | Mean | SD | Estimate | SE | t | P |
|---|---|---|---|---|---|---|
| **Owner age** | - | - | -.128 | .036 | -3.57 | .0004 |
| **Owner gender** | | | - | - | - | - |
| Female (ref) | -.112 | .824 | - | - | - | - |
| Male | .114 | 1.08 | .185 | .068 | 2.69 | .007 |
| **Dog source** | | | | | | |
| Purebred breeder (ref) | .0524 | 1.03 | - | - | - | - |
| Friend/relative/neighbour | .108 | 1.05 | .252 | .128 | 1.97 | .049 |
| **Puppy background** | | | | | | |
| Canadian breeder (ref) | .122 | 1.06 | - | - | - | - |
| Shelter/rescue/vet clinic | -.121 | .940 | -.317 | .159 | -1.99 | .047 |
| Stranger's dog | -.206 | .730 | -.339 | .149 | -2.267 | .024 |
| Unknown | -.301 | .732 | -.426 | .125 | -3.397 | .0007 |

CBARQ Factor 1 "Difficult behaviour" multivariate regression results. Mean and SD are provided for categorical variables.

Additionally, owners of dogs that were acquired from a pet store compared to dogs obtained from a purebred breeder, and owners of dogs that were born to a dog from an international breeder compared to dogs that were born to a dog from Canadian breeder scored higher for the "Struggle" factor.

Increase in owner age predicted a lower score for the "Burden" factor (Table 7). Increase in the number of children in the household predicted an increase in the "Burden" score. Owners with university, and postgraduate education compared to owners who reported "Primary school/Secondary school" as their highest level of education had higher "Burden" scores.

Dog source and puppy background revealed confusing results on the "Burden" factor score. Given that we separated dog source and puppy background into two measures with slight differences in the phrasing and nuance of the questions asked, interpretations may have varied between respondents, resulting in reduced interpretability on the current findings. The current study found that owners of dogs that were acquired from another dog in the household had reduced "Burden" scores than those that were acquired from a purebred breeder. Confusingly, owners of dogs that were born in the household had higher "Burden" scores compared to dogs that were born to a dog from Canadian breeder.

Owners of Canadian dogs scored higher for the "Regret" factor score than owners of non-Canadian dogs (Table 8). Increase in owner age predicted lower "Regret" scores. Increase in the medical expenses spent predicted higher "Regret" scores. Owners with high school or trade school, university, or a postgraduate education as the highest level of education compared to owners with "Primary school/Secondary school" for their highest level of education received,

**Table 5. Statistically significant interactions of owner and dog parameters on CBARQ Factor 2 "Excitability".**

| Parameter | Mean | SD | Estimate | SE | t | P |
|---|---|---|---|---|---|---|
| **Dog age** | - | - | -.043 | .008 | -5.231 | 2.17e-7 |
| **Puppy background** | | | | | | |
| Canadian breeder (ref) | .122 | 1.06 | - | - | - | - |
| International breeder | .106 | .990 | -.576 | .198 | -2.905 | .004 |

CBARQ Factor 2 "Excitability" multivariate regression results. Mean and SD are provided for categorical variables.

**Table 6. Statistically significant interactions of owner and dog parameters on MDORS Factor 1 "Struggle".**

| Parameter | Mean | SD | Estimate | SE | t | P |
|---|---|---|---|---|---|---|
| **Dog origin** | | | | | | |
| Non-Canadian (ref) | -.069 | .911 | - | - | - | - |
| Canadian | .005 | .937 | .448 | .205 | 2.181 | .029 |
| **Owner age** | - | - | -.124 | .034 | -3.609 | .0003 |
| **Owner education** | | | | | | |
| Primary/Secondary (ref) | -.333 | .889 | - | - | - | - |
| High/Trade | -.126 | .905 | .282 | .138 | 2.053 | .04 |
| University | .132 | .896 | .465 | .138 | 3.359 | .0008 |
| Postgraduate | .254 | 1.08 | .563 | .162 | 3.486 | .0005 |
| **Owner gender** | | | | | | |
| Female (ref) | -.149 | 849 | - | - | - | - |
| Male | .162 | .996 | .262 | .065 | 4.015 | 6.53e-5 |
| **Dog source** | | | | | | |
| Purebred breeder (ref) | .010 | .948 | - | - | - | - |
| Pet store | .558 | 1.06 | .470 | .188 | 2.498 | .013 |
| **Puppy background** | | | | | | |
| Canadian breeder (ref) | .009 | .955 | - | - | - | - |
| International breeder | .455 | 1.30 | .601 | .202 | 3.016 | .003 |

MDORS Factor 1 "Struggle" multivariate regression results. Mean and SD are provided for categorical variables.

and male owners compared to female owners had higher "Regret" scores. Additionally, owners who acquired their dog from a pet store had higher "Regret" scores compared to owners who acquired their dog from a purebred breeder. Owners of dogs that were born to a dog from an international breeder also had higher "Regret" scores than owners with dogs that were born to a dog from a Canadian breeder. Confusingly, the "household as source" response category for dog source and puppy background had opposite effects on the "Regret" score. Owners who acquired their dog from another dog in their household had lower "Regret" scores than owners

**Table 7. Statistically significant interactions of owner and dog parameters on MDORS Factor 2 "Burden".**

| Parameter | Mean | SD | Estimate | SE | t | P |
|---|---|---|---|---|---|---|
| **Owner age** | - | - | -.136 | .0328 | -4.138 | 3.89e-5 |
| **Children in household** | - | - | .075 | .0292 | 2.566 | .011 |
| **Owner education** | | | | | | |
| Primary/Secondary (ref) | -.0759 | 1.08 | - | - | - | - |
| University | .097 | .887 | .259 | .132 | 1.966 | .049 |
| Postgraduate | .305 | 1.06 | .478 | .154 | 3.103 | .002 |
| **Dog source** | | | | | | |
| Purebred breeder (ref) | .117 | 1.02 | - | - | - | - |
| Household offspring | -.194 | .597 | -.591 | .267 | -2.171 | .030 |
| **Puppy background** | | | | | | |
| Canadian breeder (ref) | .0639 | .950 | - | - | - | - |
| Household offspring | .0108 | .886 | .496 | .247 | 2.007 | .045 |

MDORS Factor 2 "Burden" multivariate regression results. Mean and SD are provided for categorical variables.

**Table 8. Statistically significant interactions of owner and dog parameters on HAB Factor 1 "Regret".**

| Parameter | Mean | SD | Estimate | SE | t | P |
|---|---|---|---|---|---|---|
| **Dog origin** | | | | | | |
| Non-Canadian (ref) | -.112 | .767 | - | - | - | - |
| Canadian | .009 | .967 | .419 | .202 | 2.075 | .038 |
| **Owner age** | - | - | -.161 | .034 | -4.74 | 2.54e-6 |
| **Medical cost** | - | - | 2.181e-5 | 8.075e-6 | 2.701 | .007 |
| **Owner education** | | | | | | |
| Primary/Secondary (ref) | .366 | .730 | - | - | - | - |
| High/Trade | -.122 | .839 | .381 | .135 | 2.819 | .004 |
| University | .096 | .984 | .492 | .136 | 3.618 | .0003 |
| Postgraduate | .382 | 1.22 | .754 | .159 | 4.749 | 2.44e-6 |
| **Owner gender** | | | | | | |
| Female (ref) | -.149 | .848 | - | - | - | - |
| Male | .158 | 1.03 | .228 | .064 | 3.566 | .0004 |
| **Dog source** | | | | | | |
| Purebred breeder (ref) | .032 | .978 | - | - | - | - |
| Household offspring | -.335 | .605 | -.542 | .275 | -1.97 | .049 |
| Pet store | .920 | 1.22 | .765 | .185 | 4.132 | 3.99e-5 |
| **Puppy background** | | | | | | |
| Canadian breeder | .032 | .990 | - | - | - | - |
| International breeder | .606 | 1.42 | .718 | .198 | 3.619 | .0003 |
| Household offspring | .180 | 1.16 | .578 | .255 | 2.269 | .024 |

HAB Factor 1 "Regret" multivariate regression results. Mean and SD are provided for categorical variables.

who got their dog from a purebred breeder. Owners of dogs that were born in their household had higher "Regret" scores than those whose dog came from from a Canadian breeder.

Increase in the number of children in the household predicted a decrease in the "Satisfaction" factor score (Table 9). Male owners had lower for the "Satisfaction" scores compared to female owners. Additionally, owners with dogs that were born to another dog in the household also had a lower "Satisfaction" score compared to owners who acquired their dog from a purebred breeder.

Owners of Canadian dogs had higher "Harsh training" factor scores than owners of non-Canadian dogs (Table 10). Surprisingly, owners of international rescue dogs had higher "Harsh training" scores than owners of non-international rescue dogs. Increase in owner age

**Table 9. Statistically significant interactions of owner and dog parameters on HAB Factor 2 "Satisfaction".**

| Parameter | Mean | SD | Estimate | SE | t | P |
|---|---|---|---|---|---|---|
| **Children in household** | - | - | -.089 | .031 | -2.870 | .004 |
| **Owner gender** | | | | | | |
| Female (ref) | .125 | .839 | - | - | - | - |
| Male | -.135 | .957 | -.236 | .066 | -3.603 | .0003 |
| **Dog source** | | | | | | |
| Purebred breeder (ref) | .032 | .978 | - | - | - | - |
| Household offspring | -.335 | .605 | -.660 | .282 | -2.345 | .019 |

HAB Factor 2 "Satisfaction" multivariate regression results. Mean and SD are provided for categorical variables.

**Table 10. Statistically significant interactions of owner and dog parameters on training methods Factor 1 "Harsh training".**

| Parameter | Mean | SD | Estimate | SE | t | P |
|---|---|---|---|---|---|---|
| **Dog origin** | | | | | | |
| Non-Canadian (ref) | -.331 | .642 | - | - | - | - |
| Canadian | .026 | .927 | .717 | .188 | 3.827 | .0001 |
| **International rescue** | | | | | | |
| No (ref) | .016 | .922 | - | - | - | - |
| Yes | -.400 | .558 | .534 | .243 | 2.200 | .028 |
| **Owner age** | - | - | -.120 | .031 | -3.813 | .0002 |
| **Dog age** | - | - | -.020 | .008 | -2.601 | .009 |
| **Children in household** | - | - | .09 | .028 | 3.218 | .001 |
| **Medical cost** | - | - | 1.547e-5 | 7.505e-6 | 2.061 | .039 |
| **Owner education** | | | | | | |
| Primary/Secondary (ref) | .044 | 1.03 | - | - | - | - |
| Postgraduate | .298 | 1.08 | .324 | .148 | 2.199 | .028 |
| **Owner gender** | | | | | | |
| Female (ref) | -.176 | .784 | - | - | - | - |
| Male | .186 | .999 | .263 | .059 | 4.423 | 1.11e-5 |
| **Dog source** | | | | | | |
| Purebred breeder (ref) | .097 | .972 | - | - | - | - |
| Pet store | .914 | 1.18 | .803 | .172 | 4.669 | 3.56e-6 |
| **Puppy background** | | | | | | |
| Canadian breeder (ref) | .126 | .976 | - | - | - | - |
| Stranger's dog | -.419 | .414 | -.374 | .130 | -2.867 | .004 |
| International breeder | .399 | .131 | .514 | .184 | 2.787 | .006 |
| Unknown | -.362 | .617 | -.267 | .109 | -2.444 | .015 |

Training methods Factor 1 "Harsh training" multivariate regression results. Mean and SD are provided for categorical variables.

and dog age predicted a decrease in the "Harsh training" score. Increase in the number of children in the household, as well as increase in medical expenses spent predicted increase in the "Harsh training" score. Owners with postgraduate education compared to owners with primary or secondary school education as their highest education, and male owners compared to female owners scored higher "Harsh training" scores. Additionally, owners that acquired their dog from a pet store had higher "Harsh training" scores compared to those that acquired their dog from a purebred breeder. Owners of dogs that were born to a dog from an international breeder also scored higher "Harsh training" scores compared to those that were born to a dog from a Canadian breeder. Owners with dogs that were born to a stranger's dog, and owners with dogs with unknown background as a puppy scored lower on the "Harsh training" factor score compared to owners with dogs that were born a dog from a Canadian breeder.

Increase in dog age predicted a decrease in "Gentle training" score (Table 11). The "household as source" category for dog source and puppy background had opposite effect on the "Gentle training" factor score. Dogs that were acquired from another dog in the household had higher "Gentle training" score than dogs that were acquired from a purebred breeder. Confusingly, dogs that born to another dog in the household had lower "Gentle training" score than dogs that born to a dog from a Canadian breeder.

Owners of Canadian dogs had higher "Perceived health issues" factor scores than owners of non-Canadian dogs (Table 12). Increase in owner age predicted lower "Perceived health

**Table 11. Statistically significant interactions of owner and dog parameters on training methods Factor 2 "Gentle training".**

| Parameter | Mean | SD | Estimate | SE | t | P |
|---|---|---|---|---|---|---|
| **Dog age** | - | - | -.034 | .007 | -4.675 | 3.47e-6 |
| **Dog source** | | | | | | |
| Purebred breeder (ref) | .029 | .787 | - | - | - | - |
| Household offspring | .138 | .437 | .522 | .239 | 2.178 | .029 |
| **Puppy background** | | | | | | |
| Canadian breeder (ref) | .018 | .813 | - | - | - | - |
| Household offspring | -.336 | 1.02 | -.646 | .222 | -2.908 | .004 |

Training methods Factor 2 "Gentle training" multivariate regression results. Mean and SD are provided for categorical variables.

issues" scores. Increase in dog age, as well as the number of children in the household predicted higher "Perceived health issues" scores. Dogs that were acquired from a pet store had higher "Perceived health issues" scores compared to those that were acquired from a purebred breeder. Interestingly, dogs that originally came from a pet store as a puppy had lower "Perceived health issues" score than those that born to a dog from a Canadian breeder. Dogs that were born to a dog from an international breeder had higher "Perceived health issues" compared to those that were born to a dog from a Canadian breeder.

Owners of Canadian had higher "Expectation" factor score than owners of non-Canadian dogs (Table 13). Increase in owner age and dog age predicted lower "Expectation" scores. Dogs that were acquired from another dog in the household had lower "Expectation" scores compared to those that were acquired from a purebred breeder. Additionally, dogs that were born to a stranger had lower "Expectation" scores compared to dogs that were born to a dog from a Canadian breeder. Dogs that were born to a dog from an international breeder, and dogs that were born from another dog in the household had higher "Expectation" scores than those that were born to a dog from a Canadian breeder.

Increase in dog age, and the number of children in the household predicted a lower "Attachment" factor score (Table 14). Male owners had lower "Attachment" score than female owners.

**Table 12. Statistically significant interactions of owner and dog parameters on perceived health factor "Perceived health issues".**

| Parameter | Mean | SD | Estimate | SE | t | P |
|---|---|---|---|---|---|---|
| **Dog origin** | | | | | | |
| Non-Canadian (ref) | -.209 | .557 | - | - | - | - |
| Canadian | .016 | .973 | .418 | .208 | 2.007 | .045 |
| **Owner age** | - | - | -.071 | .035 | -2.04 | .042 |
| **Dog age** | - | - | .053 | .009 | 6.208 | 8.74e-10 |
| **Children in household** | - | - | .076 | .031 | 2.442 | .015 |
| **Dog source** | | | | | | |
| Purebred breeder (ref) | .141 | 1.05 | - | - | - | - |
| Pet store | .674 | 1.37 | .645 | .191 | 3.377 | 7.68e-4 |
| **Puppy background** | | | | | | |
| Canadian breeder (ref) | .091 | 1.03 | - | - | - | - |
| International breeder | .331 | 1.03 | .413 | .205 | 2.015 | .044 |
| Pet store | .189 | 1.18 | -.509 | .251 | -2.026 | .043 |

Perceived health Factor "Perceived health issues" multivariate regression results. Mean and SD are provided for categorical variables.

**Table 13. Statistically significant interactions of owner and dog parameters on expectation factor "Expectation".**

| Parameter | Mean | SD | Estimate | SE | t | P |
|---|---|---|---|---|---|---|
| **Dog origin** | | | | | | |
| Non-Canadian (ref) | -.358 | .693 | - | - | - | - |
| Canadian | .028 | .843 | .477 | .183 | 2.601 | .009 |
| **Owner age** | - | - | -.116 | .0308 | -3.759 | .0002 |
| **Dog age** | - | - | -.027 | .008 | -3.579 | .0004 |
| **Dog source** | | | | | | |
| Purebred breeder (ref) | .029 | .878 | - | - | - | - |
| Household offspring | -.289 | .795 | -.543 | .250 | -2.174 | .030 |
| **Puppy background** | | | | | | |
| Canadian breeder (ref) | .040 | 864 | - | - | - | - |
| Stranger's dog | -.408 | .706 | -.407 | .128 | -3.191 | .0015 |
| International breeder | .266 | .931 | .452 | .180 | 2.507 | .012 |
| Household Offspring | .098 | .900 | .478 | .232 | 2.064 | .039 |

Expectation Factor "Expectation" multivariate regression results. Mean and SD are provided for categorical variables.

**Dog care responses as a function of owner and dog parameters.** *Vet visit*. The logistic regression model for the Vet visit question obtained a statistically significant intercept value (Estimate = 4.120, SE = 1.586, z = 2.598, P = .0094). However, none of the predictor variables obtained a statistically significant value. The predictor variable that had the most significance was the "Born to dog living on street/found/stray" category for the puppy background question (Estimate = -1.35, SE = .783, z = -1.725, P = .085).

*Professional help*. Owner age and dog age were strong predictors of the Professional help question response, where both older owners and owners of older dogs were more likely to seek help from a professional trainer (Owner age: Estimate = .216, SE = .083, z = 2.602, P = .009; Dog age: Estimate = .106, SE = .021, z = 4.892, P < .001) (Fig 1). Gender was also a strong predictor of the outcome for the Professional help question, where male owners had decreased likelihood of seeking help from a professional trainer compared to female owners (Estimate = -.365, SE = .157, z = -2.330, P = .020).

*Internet help*. Owner age and dog age were strong predictors of the Internet help question outcome, where older owners and owners of older dogs were more likely to access online resources when training their dog (Owner age: Estimate = .377, SE = .098, z = 3.864 P < .001; Dog age: Estimate = .121, SE = .022, z = 5.475, P < .001) (Fig 2). Additionally, owners with university or postgraduate education were less likely, compared to owners with primary school/secondary school as their highest level of education, to access online resources (University:

**Table 14. Statistically significant interactions of owner and dog parameters on LAPS factor "Attachment".**

| Parameter | Mean | SD | Estimate | SE | t | P |
|---|---|---|---|---|---|---|
| **Dog age** | - | - | -.019 | .009 | -2.064 | .039 |
| **Children in household** | - | - | -.104 | .034 | -3.158 | .002 |
| **Owner gender** | | | | | | |
| Female (ref) | .183 | .909 | - | - | - | - |
| Male | -.190 | 1.00 | -.352 | .069 | -5.03 | 6.08e-7 |

LAPS Factor "Attachment" multivariate regression results. Mean and SD are provided for categorical variables.

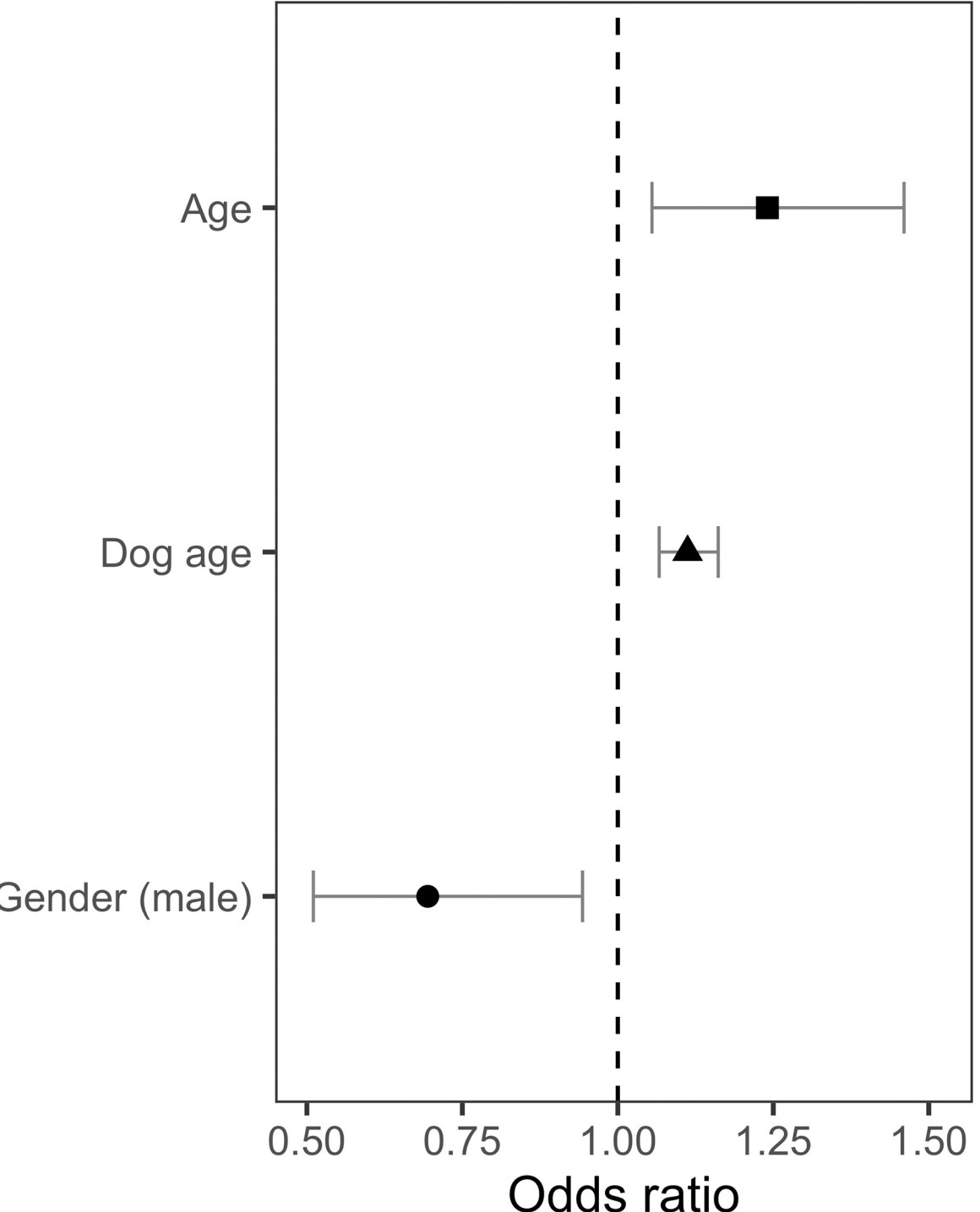

**Fig 1. Statistically significant owner and dog parameters on the professional help outcome.** Professional help outcome odds predicted by statistically significant predictor variables: owner age (P = .009), dog age (P < .001), and male owners (P = .019), calculated by the logistic regression model.

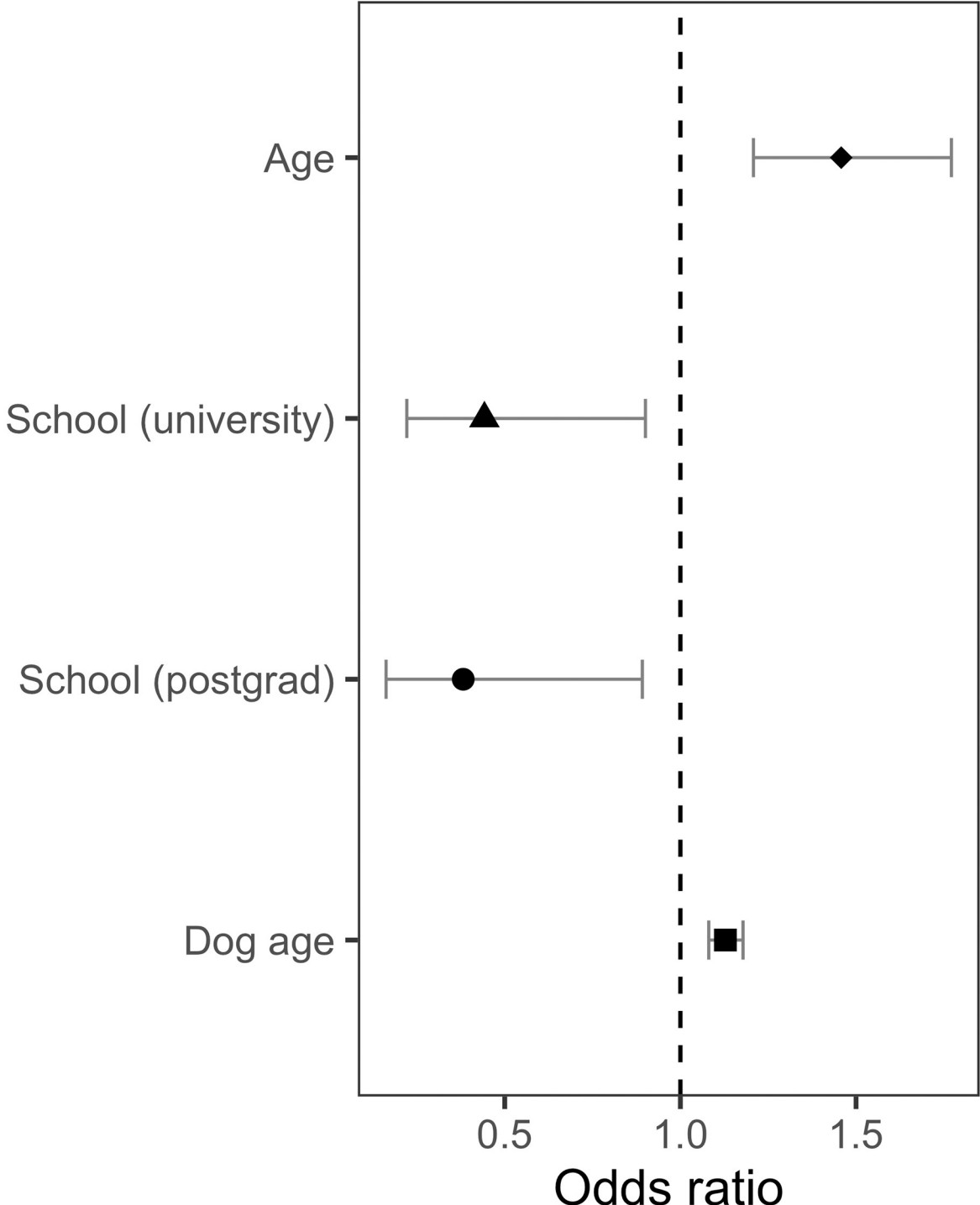

**Fig 2. Statistically significant owner and dog parameters on the internet help outcome.** Internet help outcome odds predicted by statistically significant predictor variables: owner age (P < .001), university-educated owners (P = .022), postgraduate-educated owners (P = .026), and dog age (P < .001), calculated by the logistic regression model.

Estimate = -.816, SE = .356, z = -2.290, P = .022; Postgraduate: Estimate = -.963, SE = .432, z = -2.224, P = .026).

*Experience help.* Owner age was a strong predictor of the Experience help question outcome. Interestingly, younger owners were more likely to indicate that they used their previous experience with dogs when training their dog compared to older owners (Estimate = -.461, SE = .117, z = -3.947, P < .001) (Fig 3). The number of children in the household was also a strong predictor of the Experience help question outcome, where households with more children were more likely to rely on previous experience (Estimate = .192, SE = .095, z = 2.025, P = .043).

## Discussion

**Effect of dog origin.** The objective of this study was to investigate whether the origin of dogs (Canadian or from abroad) influenced the owner-dog relationship. Imported dogs come from various backgrounds, which in some cases may result in difficult behaviours, posing challenges for some dog owners [42, 43]. However, the prevalence of these issues in Canada is currently unknown. Since owners are the primary caretakers of dogs, and, ultimately make decisions about the dog's future, owner-reported differences were used to assess whether imported dogs are at a greater risk for a dysfunctional owner-dog relationship, which may result in relinquishment in extreme cases [33, 44].

The current study found that owners of Canadian and non-Canadian dogs did not report differences in the level of difficult behaviours displayed by their dog, satisfaction with their dog, or the level of attachment to their dog. Interestingly, owners of Canadian dogs reported to struggle more with their dog, have more regrets, perceive more health issues, use harsher training methods more frequently, and have higher expectations for their dog.

A possible explanation for the findings observed in the current study may be due to differences between owners that chose to acquire dogs domestically versus internationally. Half of owners of non-Canadian dogs acquired their dog through a shelter, rescue, or a veterinary clinic whereas roughly only a quarter of owners of Canadian dogs chose to acquire their dog through this method. These differences may indicate greater altruistic motives for those who acquire dogs internationally. Differences in motives for acquiring a dog may also influence the expected role of the dog in the household, seen from the differences in the Expectation factor. It is also plausible that owners with altruistic motives may also show greater concern for the well-being of animals compared to owners with other motives.

However, further research is needed to understand why such differences were found. Moreover, due to the small sample size and low Cronbach alpha score for the Expectation factor, generalizations of these results should be done with caution. Nonetheless, findings of the current study go against the findings of Munkeboe et al. (2021), where owners in Denmark reported more difficult behaviours for imported dogs than domestically reared dogs. Furthermore, their data showed that compared to owners, a greater proportion (86%) of veterinarians reported seeing more behavioural problems in imported dogs compared to domestic dogs. The conflicting findings of this study highlight a need for further research to understand how, if at all, imported and domestically-sourced dogs differ, and subsequently influence the owner-dog relationship.

**Pet store as source.** Previous literature suggests dogs coming from pet stores to be at a greater risk of developing problematic behaviours compared to dogs obtained from other sources [29, 30, 45]. Aggression is among the most commonly reported problem behaviour occurring in dogs acquired from pet stores [28]. Interestingly, our data did not reveal differences in difficult behaviours for dogs originating from pet stores compared to dogs from other

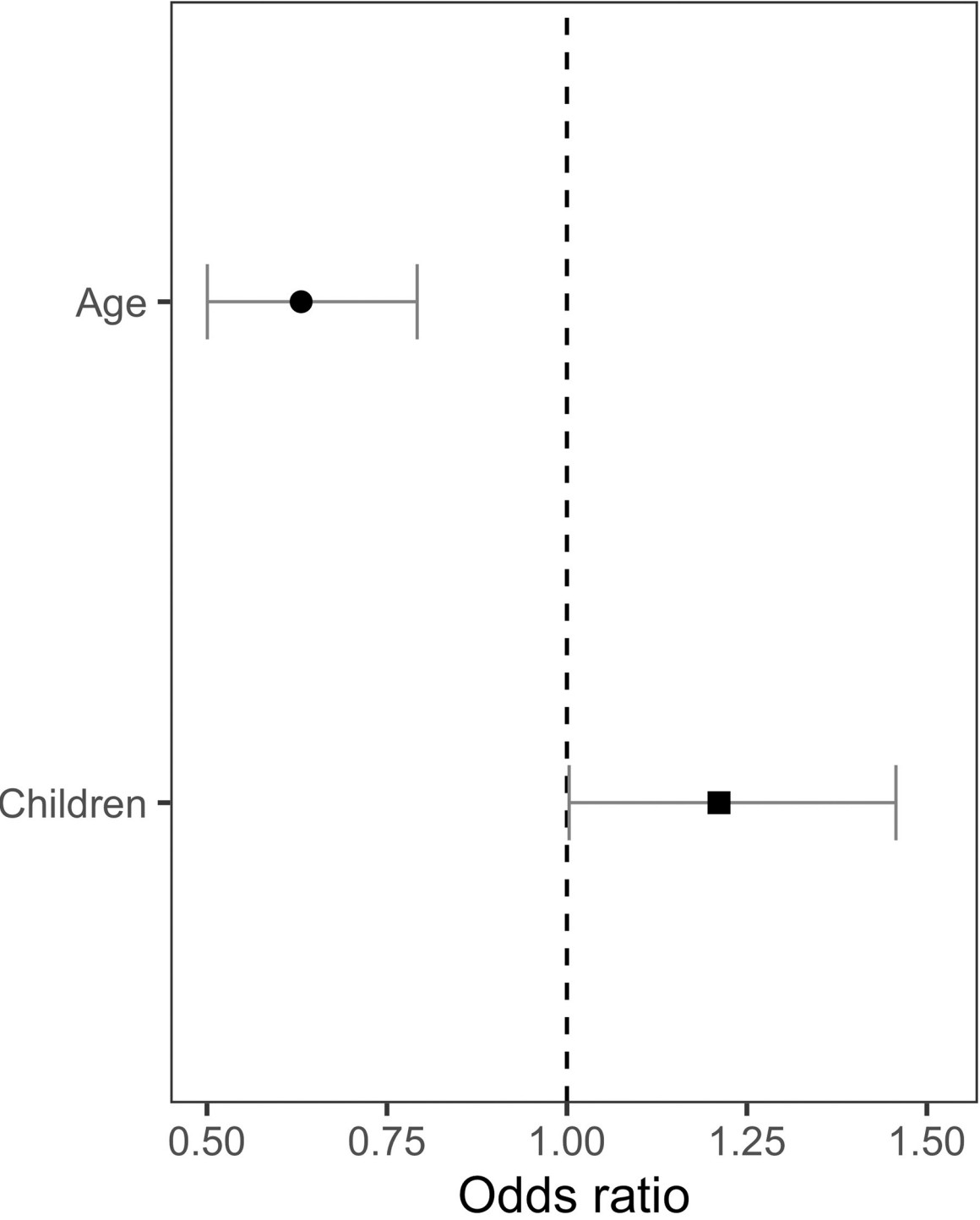

**Fig 3. Statistically significant owner and dog parameters on experience help outcome.** Experience help outcome odds predicted by statistically significant predictor variables: owner age (P < .001) and the number of children in the household (P = .043), calculated by the logistic regression model.

sources. However, owners of pet store-sourced dogs reported an increased use of harsher training methods, increased perceived health issues, increased levels of struggle, and had more regrets compared to owners who acquired their dog from a purebred breeder. However, interpretations of these results should be done with caution due to the small sample size of dogs that were obtained from pet stores (n = 41).

Nevertheless, the greater struggle and regret experienced by owners of pet store-sourced dogs highlight a potentially significant risk to the owner-dog relationship. Additionally, the increased use of harsh training methods and increased perceived health issues suggest a possible risk to dog welfare. Although the current study did not reveal the causality of these dimensions, these findings support the mounting evidence that pet stores, or large-scale commercial dog breeding facilities that often supply pet stores, do not promote dog welfare [28–30].

**Source as a puppy.**   Our data revealed that owners of dogs that were originally bred by an international purebred breeder reported increased use of harsher training methods, increased expectations for their dog, perceived more health issues, and experienced increased levels of struggle and regret, but decreased excitability compared to puppies sourced from a Canadian breeder.

Unrealistic owner expectations have been found to negatively influence the owner-dog relationship [46–48]. Accordingly, it is possible that owners of puppies obtained from an international purebred breeder devoted more time, effort, and money than other owners in search of their 'ideal' pet; thus, they also set unrealistic expectations for their dogs. However, the low reliability score for the Expectation factor and small sample size for owners of puppies from international purebred breeders (n = 29) warrant further investigation into this dimension for a clearer interpretation of these findings. Nevertheless, educational efforts could be targeted at owners during the pre-acquisition period. This may help prospective owners to set more realistic expectations for the behaviours of their pets and the level of effort required to care for them, which may minimize risk for relinquishment as a result of a mismatch between owner expectations and actual caretaking efforts [3, 47, 49, 50].

It is also plausible that some proportion of dogs that were obtained from an international breeder, as reported by the owner, may be coming from large-scale commercial breeding facilities [21, 51]. Dogs that are born at these facilities may have poor early-life experiences, and consequently have a greater likelihood of displaying difficult behaviours in adulthood [28, 30, 52].

**Household as source.**   Dogs born in the household presented puzzling results. The confusion is partly due to the design of the study, where two similar, yet distinct measures of household as a source were collected. The two household as source measures were from dog source (from where the owner obtained their dog), and from puppy background (from where the dog came as a puppy). Additionally, two different reference variables were selected for the two questions (dog source: purebred breeder; puppy background: Canadian breeder), which further convoluted interpretations.

Owners with dogs acquired from another dog in the household, felt less burdened, had less regrets, felt less satisfied, used gentle training methods more frequently, and had lower expectations for their dog compared to owners with dogs acquired from a purebred breeder. Interestingly, owners with dogs that born to another dog in the household felt more burdened, had more regrets, used gentle training methods more frequently, and had higher expectations for their dog compared to owners with dogs born to a Canadian breeder. To the authors' knowledge, no prior research investigated owner-reported differences between dogs born in the household to those acquired from a breeder.

**Effect of owner and dog parameters.**   Owner age, highest level of education reached, gender, and dog age repeatedly appeared as statistically significant predictors for various owner-

dog relationship dimensions, which supports the notion that human factors greatly influence the owner-dog relationship, and possibly even more than dog factors [36]. However, it is important to note that the current study did not extensively examine dog factors such as breed, personality traits, or dog behaviour, and their effect on the current findings are unknown. Additionally, certain dog behaviours may be perceived as 'acceptable' to one owner but perceived as a 'problem' to another [47]. Therefore, it is also possible that both dog and human factors are equally important, and a good match between the dog's characteristics and human desires are necessary for a successful owner-dog relationship [47, 53]. Regardless, these findings provide further insight into owner and dog factors and their association to the owner-dog relationship.

With increase in owner age, owners reported their dogs having fewer behavioural problems, struggling less, feeling less burdened, having fewer regrets, using harsh training methods less frequently, fewer health issues, and having lower expectations of their dog. Additionally, the increase in owner age increased the odds of owners receiving help from a professional trainer or the internet, and reduced odds of relying on previous experience when training their dog. Overall, our results suggest that an increase in owner age improves the owner-dog relationship.

Although our results alone do not deny the possibility that older owners are generally 'better' dog owners, it is possible that these findings are confounded by other variables. One explanation may be the greater financial freedom that older owners may have, which may allow them greater access to resources to enhance their owner-dog relationship. This hypothesis is further supported by the increased odds of older owners accessing professional or online resources, which may have been effective at reducing the occurrence of problematic behaviours and mitigating the inconveniences of dog ownership. In support of this explanation, previous studies have found that dogs who regularly attended obedience classes had improved behaviours compared to those that did not [45, 54, 55]. Alternatively, there may have been more inherent causes of bias in the results, such as differences in attitudes. Bir et al. (2017) found owner age-related differences in attitudes regarding the way people acquired their dogs. In their study, a higher percentage of older owners (aged 55–88) agreed with statements supportive of purchasing purebred dogs and statements claiming importation of dogs even for adoption as irresponsible. This may have led owners of varying age groups to obtain their dogs through different methods or to obtain different dog breeds, further complicating the effect of age on the owner-dog relationship. Furthermore, previous literature has found no concrete effect of owner age on relinquishment [33, 47], dog's quality of life [35], and the owner-dog relationship [36], suggesting that owner age is likely confounded by other factors in its effect on the owner-dog relationship.

The current study found that highly educated owners reported poorer owner-dog relationship outcomes. Specifically, owners who reported "Highschool/Trade school", "University", and "Postgraduate" degree as their highest level of education received experienced more struggles with their dog and had more regrets than owners with primary/secondary education. Additionally, those with a "University" or "Postgraduate" degree felt more burdened by their dog, and owners with a "Postgraduate" degree used harsh training methods more often. These findings conflict with previous findings, where relinquishing dog owners were more likely to have education not beyond high school [47], and owners with at least a college degree had preference for dog adoption as a method for acquisition [56]. Previous research has reported mixed findings regarding the effect of the owner's level of education on the dog-owner relationship [35, 47]. This lack of consensus suggests that owner's level of education likely affects the owner-dog relationship jointly with other factors and is not solely responsible for owner-dog relationship outcomes.

The current study found that male owners experienced difficult behaviours in their dog more often, struggled more with their dog, had more regret, felt less satisfied, used harsher training methods, and felt less attached, in addition to being less likely to seek help from professional trainers when training their dog. In support of this finding, male owners have also been previously reported to have an increased likelihood of dog relinquishment [47], consider their dog to be more 'disobedient' [45], and score lower on dog-human companionship dimensions compared to female owners [57]. Conflictingly, others have reported male owners to form stronger emotional bonds with their dog [58] and have no difference in strength of attachment to their dog compared to female owners [59]. The findings of the current study indicate male owners to be at greater risk for poorer owner-dog relationship outcomes. Previous studies have identified gender differences in how owners choose to acquire their dog [56], their description of an 'ideal' dog [60], and the way they interact with their dog [59]. These decision-making differences between male and female owners may be the cause for the increased hardships faced by male dog owners.

Increase in dog age resulted in owner-reported reduced excitability, increased perceived health issues, lower expectations from owners, decreased use of harsher training methods, and increased likelihood of seeking help from a professional trainer and accessing online resources during training. These findings suggest that, in the current study, owners were actively seeking assistance for challenges that may come with their aging dogs. Although increase in dog age has been suggested to be correlated with unfriendliness and aggressiveness [45], no such interactions were found in our study. However, it is important to consider that behavioural problems in the current study were measured using owner report. Neighbours or other individuals in the community may perceive the same dog behaviour differently [54]. Additionally, veterinarians also perceive dog behaviors differently than owners [25]. Contrary to our expectations, increase in dog age did not result in an increase in owner-reported attachment. While it is possible that owners were equally attached to dogs at any age, this may also be a result of bias caused by the online survey-based data collection method, where respondents are generally highly committed owners [45].

## Study 2

The aim of Study 2 was to add dog variables to allow for a more thorough investigation of factors influencing the owner-dog relationship. The newly added factors were dog breed, general size of the dog, and age at which the dog was acquired. Dog breed has been shown to have an influence on perceived behaviour [61], which may subsequently influence resources and methods utilized by owners, and ultimately the owner-dog relationship. Similarly, owner training engagement and interactions have been shown to vary between smaller and larger dog breeds [62]. Furthermore, there is currently little information available regarding imported dogs arriving to Canada and, to the author's knowledge, no previous studies have investigated the characteristics of these dogs. Dogs that are imported at a young age (<8 months) are of particular concern due to the potentially detrimental effects of stress during transport [29–31] and risk for zoonotic disease introduction [15], which may pose serious health and welfare concerns.

### Methods

**Survey design.**   A second survey was developed to re-examine and extend the findings of the first study; the survey consisted only of originally designed questions (S2 Appendix). The survey was distributed online using proprietary panels from September 30, 2020, to October 4, 2020, to a convenience sample consisting of 1,507 BC residents, who are currently owning, or

have owned a dog or a cat in the past 5 years. The survey was conducted by Strategic Communications Inc. (Stratcom) for the British Columbia Society for the Prevention of Cruelty to Animals. No margin of error is available for this demographic.

The survey was broadly categorized into three sections: owner demographic information, dog parameters, and owner-dog relationship dimensions. Owner demographic information included the owner's gender and age, which generated categorical and numerical data, respectively. Dog parameter questions collected information about the dog such as their origin (Canadian or non-Canadian), breed (purebred or mixed), size (small: <10kg; medium: 10-20kg; or large: >20kg), and the age at which the dog was acquired by the current owner. All dog parameter questions generated categorical data. The owner-dog relationship dimensions were simplified from Study 1 and included just one question within each category: difficult behaviour, level of burden, owner attachment, training methods, and perceived health. Each dimension was measured using the owner's level of agreement with questions pertaining to these categories. Responses were scored on a 6-point scale (0 = I don't know, 1 = Strongly disagree, 2 = Disagree, 3 = Neutral, 4 = Agree, 5 = Strongly agree).

**Recruitment and respondents.** The inclusion criteria were identical to those of Study 1: participants were required to be a current resident of BC, Canada, to own at least one dog, and to have owned that dog for at least six months. A separate set of respondents were recruited for Study 2 compared to Study 1, since the two studies utilized separate surveying platforms. However, no formal testing (such as IP testing) was done to confirm that all respondents were unique to each study. Each participant was reimbursed for completing the survey according to their panel payment schedule. The survey was approved by the UBC Behavioural Research Ethics Board (H20-03637).

**Statistical analysis.** All data were handled and analyzed using R version 4.0.1. The original dataset and R code used in the analysis can be found in the (S2 Dataset and S1 Code). A panel of 1,507 respondents were recruited. However, recruitment for Study 2 was done jointly for a separate study examining cat owner attitudes. This resulted in only 954 participants (63%) meeting the inclusion criteria as many respondents were cat owners. From the 954 responses, certain categories were removed due to small sample sizes or missing values. This included the "non-binary" and "other terms" category for owner gender, and the "I don't know" category for dog origin, dog breed, age when acquired, and the owner-dog relationship dimensions. This resulted in 878 responses that were included in the analysis. Further details on the survey response rates were not available as recruitment was managed by an outside firm.

A logistic regression model was used to test the variables that were collected in the current study for their predictive power in the dog origin outcome. Accordingly, dog origin was entered as the outcome variable in the logistic regression model. The model's goodness of fit was tested using the Hosmer-Lemeshow test, which yielded a satisfactory outcome (Chi-squared = 3.66, df = 8, P = 0.89).

## Results

**Descriptive statistics.** The majority of the respondents were female (59.5%, n = 522) and were in the age group "55 years or older" (41.1%, n = 361). The remaining respondent ages were distributed between the age groups "35 to 54" (31.3%, n = 277) and "18 to 34" (27.3%, n = 240). Most were owners of Canadian dogs (86.9%, n = 763), and owned small-sized breeds (35.4%, n = 311). Purebred dogs were more common than mixed breeds (54.6%, n = 479), and most dogs were acquired as a puppy (53.9%, n = 473; S2 Appendix).

A higher proportion of owners of non-Canadian dogs were female (69.6%, n = 80) and were in the age group "55 years or older" (40.0%, n = 46). The majority of non-Canadian dogs

**Table 15. Canadian and non-Canadian dog sizes.**

| Dog size | Count | Percentage |
|---|---|---|
| **Canadian dog size** | **763** | **86.9** |
| Small (<10kg) | 261 | 34.21 |
| Medium (10-20kg) | 265 | 34.73 |
| Large (>20kg) | 237 | 31.06 |
| **Non-Canadian dog size** | **115** | **13.1** |
| Small (<10kg) | 50 | 43.48 |
| Medium (10-20kg) | 41 | 35.65 |
| Large (>20kg) | 24 | 20.87 |
| **Total** | **878** | **100** |

Number (count) and percentage of Canadian and non-Canadian dogs' sizes (owners were asked to estimated adult weight of a puppy).

were small dogs (43.4%, n = 50) (Table 15). Over half of non-Canadian dogs were mixed breed (Table 16). Toy breeds were the most common breed type of non-Canadian purebred dogs (34%, n = 17; S2 File). Non-Canadian dogs were mostly acquired either as a puppy (33%, n = 38) or as an adult (33%, n = 38) (Table 17). Surprisingly, there were 16 non-Canadian dogs that were acquired as a "Newborn" (<8 weeks) (13.9%, n = 16; S2 File). The majority of non-Canadian dogs were acquired through a rescue organization or a shelter (Table 18).

**Logistic regression.** Female owners, dogs acquired as an adolescent or as an adult, and large dogs yielded a statistically significant value in predicting the outcome of dog origin (Table 19). Female owners were at 1.6 odds (CI 95%: 1.054–2.602) more likely of acquiring non-Canadian dogs compared to male owners (Fig 4). Canadian dog owners were 58.1% women compared to 69.8% of non-Canadian dog owners who were women. Dogs acquired as an adolescent or as an adult were also 2.1 odds (CI95%: 1.003–4.47) and 1.9 odds (CI95%: 1.034–3.87) more likely of originating outside of Canada compared to dogs acquired as a newborn respectively. Dogs acquired as an adolescent made up 7.9% of Canadian dogs, compared to 17% of non-Canadian dogs that were acquired as an adolescent. Dogs acquired as an adult made up 17.4% of Canadian dogs compared to 33.6% of non-Canadian dogs. Large dog breeds were 0.5 odds (CI95%: .294—.865) less likely than small dog breeds to be of non-Canadian origin. Canadian dogs were comprised of 31.1% large dogs compared to 20.7% of non-Canadian dogs.

**Table 16. Canadian and non-Canadian dog breeds.**

| Dog breed | Count | Percentage |
|---|---|---|
| **Canadian dog breed** | **763** | **86.9** |
| Purebred | 429 | 56.22 |
| Mixed | 334 | 43.78 |
| **Non-Canadian dog breed** | **115** | **13.1** |
| Purebred | 50 | 43.48 |
| Mixed | 65 | 56.52 |
| **Total** | **878** | **100** |

Number (count) and percentage of Canadian and non-Canadian dog breeds (purebred or mixed breed).

**Table 17. Canadian and non-Canadian dog age on acquisition.**

| Items | Count | Percentage |
|---|---|---|
| **Canadian dog age when acquired** | **763** | **86.9** |
| Newborn (<8 weeks) | 108 | 14.16 |
| Puppy (8 weeks to <5 months) | 435 | 57.01 |
| Adolescent (5months to <1 year) | 60 | 7.86 |
| Adult (1 year to <8 years) | 131 | 17.17 |
| Senior (8 years and older) | 29 | 3.8 |
| **Non-Canadian dog age when acquired** | **115** | **13.1** |
| Newborn (<8 weeks) | 16 | 13.91 |
| Puppy (8 weeks to <5 months) | 38 | 33.04 |
| Adolescent (5 months to <1 year) | 20 | 17.39 |
| Adult (1 year to <8 years) | 38 | 33.04 |
| Senior (8 years and older) | 3 | 2.61 |
| **Total** | **878** | **100** |

Number (count) and percentage of Canadian and non-Canadian dog age when acquired by the current owner.

## Discussion

Apart from a number of owner demographic information and dog parameters, Study 2, yet again, did not reveal differences in the owner-dog relationship between Canadian and non-Canadian dog owners. The current finding supports the results of the first study, where we did not find that owners of imported dogs had a poorer owner-dog relationship compared to owners of dogs of Canadian origin. Our data conflict with statements cautioning against imported dogs [42, 43]. Most owners, regardless of the source of dogs, reported to have good owner-dog relationships.

The increased odds of female owners of obtaining their dog from a non-Canadian origin may be a reflection of owner gender differences in their preferred method of dog acquisition. Previous studies have suggested female owners to generally show greater concern for dogs' welfare [56, 63]. However, a critical source of bias exists for these studies including the current study, where female owners tend to be overrepresented [58].

Dogs acquired as an adolescent (5 months to <1 year) or as an adult (1 year to <8 years) had greater odds of being of non-Canadian origin compared to dogs that were acquired as a newborn (<8weeks). This is likely due to the fewer number of dogs being imported as a newborn. As of May 15, 2021, Canada has updated its import requirements for 'commercial

**Table 18. Non-Canadian dog source.**

| Items | Count | Percentage |
|---|---|---|
| Brought the dog when moving to Canada | 16 | 13.91 |
| Purchased from a foreign breeder | 20 | 17.39 |
| Worked with a rescue or shelter to adopt | 33 | 28.69 |
| Obtained from someone else who brought the dog to Canada | 25 | 21.74 |
| Not applicable since the dog was born in Canada | 6 | 5.22 |
| Other | 15 | 13.04 |
| **Total** | **115** | **100** |

Number (count) and percentage of respondents using the methods used to acquire their non-Canadian dog.

**Table 19. Logistic regression of owner and dog parameters on non-Canadian origin outcome.**

| Predictors | Estimate | SE | Z | Pr (>Z) |
|---|---|---|---|---|
| Owner gender (female) | .498 | .231 | 2.154 | .0313* |
| Owner age | -.001 | .007 | -.148 | .8821 |
| Acquired age (puppy) | -.507 | .321 | -1.580 | .1142 |
| Acquired age (adolescent) | .762 | .382 | 1.993 | .0463* |
| Acquired age (adult) | .695 | .337 | 2.061 | .0393* |
| Acquired age (senior) | -.479 | .676 | -.708 | .4788 |
| Breed (mixed) | .378 | .211 | 1.789 | .0736 |
| Dog size (medium) | -.336 | .238 | -1.411 | .1582 |
| Dog size (large) | -.672 | .273 | -2.455 | .0141* |
| Difficult behaviour | .151 | .092 | 1.638 | .1014 |
| Positive training | .028 | .118 | .238 | .8115 |
| No health issues | .064 | .091 | .707 | .4796 |
| Attachment | -.167 | .149 | -1.123 | .2615 |
| Burden | -.101 | .110 | -.913 | .3614 |

Results of the logistic regression model with owner demographic information, dog parameters, and owner-dog relationship dimensions in their predictive power for the non-Canadian origin outcome.

*Statistical significance (P<0.05) are indicated by asterisks.

purpose' dogs younger than 8 months of age. These dogs are required to have a rabies vaccination no earlier than 3 months of age. Following this, there is a 28-day waiting period before the dog can be exported to Canada, meaning that dogs should be at least 16 weeks of age at the time of export. Under the new import requirements, dogs also must be treated for internal and external parasites prior to export, and importers must acquire an import permit outlining post-import quarantine plans in the event that further inspection is necessary [6]. These added requirements will likely result in even fewer dogs arriving at ages younger than 16 weeks. However, the current study identified 16 non-Canadian dogs that were 'acquired as a newborn'. While the age at which these dogs were imported is unknown, there is ongoing discourse regarding the implications of importing puppies, and further investigation of the age of incoming dogs may highlight interesting ongoing importation dynamics.

Large dog breeds (>20kg) had reduced odds of being of non-Canadian origin compared to small dog breeds (<10kg). This may be due to higher demand for smaller breeds compared to larger breeds, or due to the complication associated with the transportation of larger dog breeds. Alternatively, this may also be confounded by other factors such as owner gender and age. Female owners have been shown to prefer smaller breeds [60], and smaller dog breeds have been shown to have generally older owners [62]. To the author's knowledge, no previous studies have examined the characteristics of incoming dogs and further research may be needed to support the current finding.

## General discussion

In recent years, there has been increasing concern over dogs arriving from non-Canadian origins. Two independent studies were conducted to investigate the owner-reported differences between Canadian and non-Canadian dogs in order to assess the risk of obtaining dogs from varying origins. The findings revealed that owner-dog relationships for non-Canadian dogs were equivalent to that for Canadian dogs; owners of non-Canadian dogs were equally satisfied

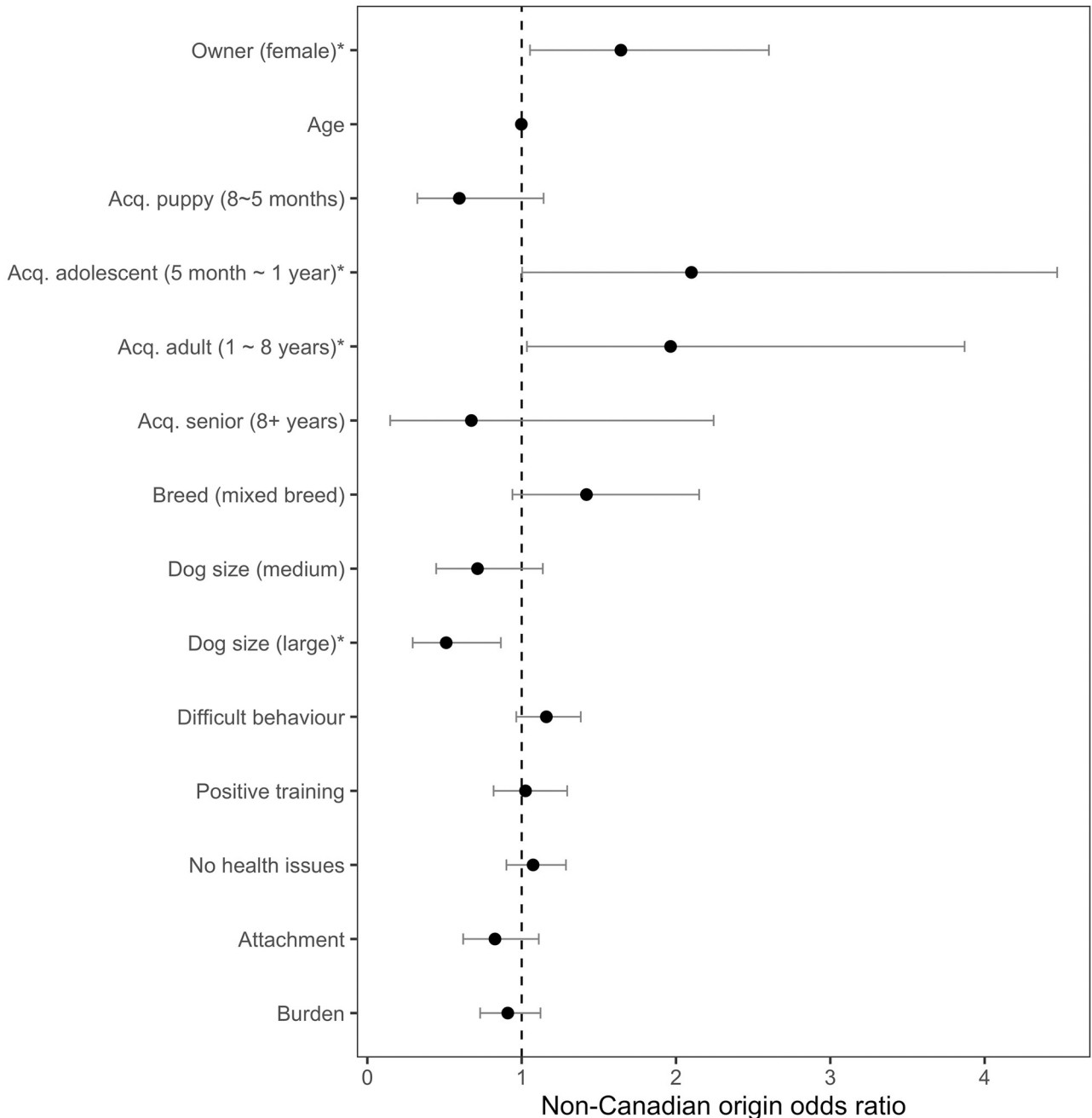

**Fig 4. Non-Canadian origin outcome odds ratio for owner and dog parameters.** Non-Canadian origin outcome odds ratio predicted by all variables collected: owner gender (female), owner age, age of dog when acquired, dog breed (mixed), dog size, and owner-dog dimensions, calculated by the logistic regression model. Statsitcal signifiance was detected when 95% confidence interval did not cross the dotted line at odds 1. Statistically significant variables are indicated with an asterisk.

with their dogs and did not report higher occurrences of problematic behaviours or health issues in their dogs compared to owners of Canadian dogs.

Study 2 revealed that 16 out of 115 non-Canadian dogs were obtained by their current owner at an age younger than 8 weeks. While the age at which these dogs were imported into

Canada is unknown, this may be of particular concern since dogs this young are not able to be vaccinated [51], which may pose a serious risk to dogs and public health [15, 16, 51]. The newly updated import requirements for dogs arriving at Canada address this issue [6]. On the other hand, this specification goes against rescue incentives, as puppies may be required to spend extended durations of time in undesirable environments before being rescued. Future studies may want to explore the balance between the needs of rescue operations and the public health risks of puppy importation.

Additionally, there has been growing reports of illegal activities where some groups may aim to profit from the growing demand for puppies [1, 2, 5, 6]. In response to the rise in illegal activities, the United States has mandated a temporary suspension on the importation of all dogs arriving from countries deemed as high-risk for canine rabies, effective from June 14, 2021 [64]. Expansion of the current permitting system to include all imported dogs (both commercial and non-commercial) may enable detailed health and background examination of incoming dogs, while also discouraging illegal activities [8]. Furthermore, this may aid in the development of effective strategy to minimize risks for zoonotic diseases that may be of particular concern.

The current finding also aligned with the growing notion that owner perceptions and attitudes may have a greater influence on the owner-dog relationship than dog characteristics [32, 34–36]. The first study found that owner-reported difficult behaviours were greatly influenced by owner parameters such as age and gender, while dog parameters such as source and age had little effect. Similarly, the second study did not find differences in owner-reported difficult behaviours between Canadian and non-Canadian dogs. Researchers suggest owner expectations to have a significant influence on the owner-dog relationship [34, 47, 48]. Owners often have expectations prior to acquiring a dog, but these expectations are not always accurate [65]. Prospective owners, especially those that have never previously owned a dog, may greatly benefit from educational efforts designed to provide a more realistic expectation of the potential challenges and behaviours they may encounter.

The second study revealed that female owners were more likely to be owners of non-Canadian dogs. Previous research has also found that female owners tend to show greater concern for dogs' welfare [56, 63], are less likely to relinquish their dog [47], are less likely to report their dog to be 'disobedient' [45], and score higher on dog-human companionship dimensions compared to male owners. Since female owners, who are more tolerant of the hardships associated with dog ownership, are also likely to acquire dogs from abroad, this may have resulted in the suppression of the effect of source on the owner-dog relationship.

## Limitations

The current study had a number of limitations, including the data collection method. All findings presented were obtained through online surveys of dog owners, which are known to create bias in the responses, as these surveys are primarily completed by highly attached and committed female dog owners [35, 45, 58]. Although participant reimbursement may have promoted survey completion from less committed owners as it was not fully volunteer-based. Furthermore, responses were limited to participants with internet access since the survey was distributed online. Responses from owners without internet access may differ from those with internet access, as owners may encounter different educational resources in their daily lives, which may influence their attitudes towards dog ownership practices.

Additionally, all data were owner-reported. Owners may not always accurately interpret dog behaviours [66]. Munkeboe et al. (2021) found dog behaviour reported by owners differed from that of veterinarians; veterinarians reported higher occurrences of difficult behaviours

for imported dogs than did owners. However, it is also plausible that difficult dog behaviours become more pronounced during a veterinary visit. Nonetheless, this highlights the potential bias in our current study where dog behaviour was evaluated solely through owner reports.

Another potential source of bias in the current study is that all survey respondents were limited to dog owners in British Columbia. This design feature reduces the applicability of the current findings to other regions in Canada, since dogs that are imported to British Columbia may have different backgrounds compared to dogs imported to other provinces. Several factors may influence dog import decisions, including local owner preferences, local rescuing activities, as well as dog rescue needs in neighbouring communities. For example, the Ontario SPCA, along with other Ontario rescue groups collaborate with First Nations communities to import dogs from northern regions of Canada through collaboration with international organizations such as the International Fund for Animal Welfare [67]. While BC rescue organizations also rescue from northern regions, collaborative efforts may not be as prominent compared to other provinces. This my influence the characteristics of dogs in British Columbia.

The analysis of the current study was exploratory in nature, which may have resulted in spurious findings. Further research is needed to confirm the findings of the current study. Similarly, the exploratory nature of the current study did not allow for the determination of causal interactions between variables and any important interaction effects [68].

In addition, the small sample size of the present study may affect the reliability of the findings. This effect is amplified for the findings regarding dogs of non-Canadian origin, as only 58 responses were obtained in Study 1 and only 115 for Study 2. Furthermore, a majority of non-Canadian dogs in Study 1 (n = 35, 60.3%) came from the United States, which may have also been the case for Study 2, although not examined. This greatly hinders our ability to generalize the findings presented to all non-Canadian dogs.

Finally, the "Gentle training" factor and "Expectation" factor for Study 1 suffered from low reliability scores (.55 and .61 respectively); any inferences that were made through these factors should be done with caution.

## Future research

Future research should continue investigation of the country of origin for dogs arriving to Canada and their background (rescued from the streets, relinquished by their previous owner, etc.). Additionally, the age of the dog at the time of importation may be a meaningful area to investigate due to a number of non-Canadian dogs that were acquired at very young age. It may also be meaningful to explore owner expectations and their effect on the owner-dog relationship as the current study did not allow for a thorough investigation. Owner expectations are not always accurate, and the owner-dog relationship may be heavily influenced as a result [46–48, 65]. Finally, investigation of the motives of owners that chose to acquire dogs from non-Canadian sources may reveal interesting sociopolitical factors influencing dog importation.

## Conclusion

The current study found no evidence of owner-reported differences in the owner-dog relationship between Canadian and non-Canadian dogs. Owners of non-Canadian dogs were equally satisfied with their dog, equally attached to their dog, and did not report higher occurrences of problematic behaviours or health issues in their dog compared to owners of Canadian dogs. However, a considerable number of non-Canadian dogs were acquired at ages younger than eight weeks. While the age at which these dogs were imported is not known, further

investigation into the age, characteristics, and backgrounds of incoming dogs is advisable. The findings of the current study also support the notion that owner attitudes may have greater influence on the owner-dog relationship than characteristics of the dog.

## Supporting information

**S1 Appendix. Study 1 full survey.**
(PDF)

**S2 Appendix. Study 2 full survey.**
(PDF)

**S1 File. Study 1 survey components and supporting tables.**
(DOCX)

**S2 File. Study 2 supporting tables.**
(DOCX)

**S1 Dataset. Study 1 original dataset.**
(XLSX)

**S2 Dataset. Study 2 original dataset.**
(XLSX)

**S1 Code. Study 1 and Study 2 R code.**
(PDF)

## Acknowledgments

We would like to thank the members of the Animal Welfare Program at the University of British Columbia for their feedback and encouragement. We would like to also thank Lexis Ly, Cheng Yu Hou, and Bailey Eagan for their assistance with R programing.

## Author Contributions

**Conceptualization:** Kai Alain von Rentzell, Karen van Haaften, Amy Morris, Alexandra Protopopova.

**Data curation:** Kai Alain von Rentzell, Alexandra Protopopova.

**Formal analysis:** Kai Alain von Rentzell, Alexandra Protopopova.

**Funding acquisition:** Alexandra Protopopova.

**Investigation:** Kai Alain von Rentzell.

**Methodology:** Kai Alain von Rentzell, Alexandra Protopopova.

**Project administration:** Kai Alain von Rentzell.

**Resources:** Alexandra Protopopova.

**Software:** Kai Alain von Rentzell.

**Supervision:** Alexandra Protopopova.

**Writing – original draft:** Kai Alain von Rentzell.

**Writing – review & editing:** Kai Alain von Rentzell, Karen van Haaften, Amy Morris, Alexandra Protopopova.

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
