## [Decision Letter · Decision Letter 0]

22 Sep 2021

PONE-D-21-25410Investigation into owner-reported differences between dogs born in versus imported into CanadaPLOS ONE

Dear Dr. von Rentzell,

Thank you for submitting your manuscript to PLOS ONE. After careful consideration, we feel that it has merit but does not fully meet PLOS ONE’s publication criteria as it currently stands. Therefore, we invite you to submit a revised version of the manuscript that addresses the points raised during the review process.

Dear Kai-

I believe that you and your co-authors have the core of a very relevant manuscript here that addresses a little-studied issue which is important to get out into the literature on human-animal relationships and animal welfare. However, as you can see from the Reviewer comments, there is substantial revision required before the manuscript can be published.

I agree with both reviewers that the data presented in the manuscript needs to be streamlined, as there are just too many data presented to the readers. I usually don’t take a hard position on this (as my general position is that, collectively, we have streamlined our academic articles far too much!). However, while your attention to detail and thoroughness is admirable, I concur with the general sentiment of the reviewers that your signal/message is getting lost in the noise/background.

I believe the suggestion of both reviewers regarding merging and/or reducing the number of data tables has a lot of merit. While you already have substantial information in your Supplementary Information, it is well worth to consider what else- currently now in the Results section of the text- belongs there, as well. As both reviewers indicate, the reader currently needs to search out the major finding for Study 1, in particular. I do not believe that this means the other exploratory analyses you conducted in Study 1 are not of interest; however, as they were exploratory, I ask you to consider how you might dramatically condense the results and present the main message(s) in a more straight-forward way.  Currently, I’d suggest that the level of detail provided in the manuscript is very appropriate for a thesis chapter, but much less so for a journal article.

I also want to note that in Study 2, you do not actually provide any statement in the Results section that indicates there were no differences found between Canadian and non-Canadian dogs/owners for the dog-owner relationship factors you assessed.

I believe that both studies can be reported in the same manuscript, as long as the material reported for Study 1 is reduced. I do want you to consider the comment from Reviewer 1 regarding whether you can justly claim that Study 2 ‘replicates’ Study 1. Certainly, the findings are consistent with the outcomes of Study 1, but particularly as you haven’t shown what happened with the dog-owner relationships dimensions in Study 2, I am not sure using the term ‘replicate’ is fair at this point. While this might be semantic, re-visiting the relationship between Study 1 and 2 might help clarify for the reader the different contribution that each study is making regarding the possible differences between who owns imported dogs in Canada, and whether there are differences in the reported behavioural problems and owner-dog relationships between these groups.

Reviewer 1 suggests that the CHERRIES reporting guidelines for online surveys might be useful to you. I admit not knowing these, so I looked them up. You can find more information here in this article (as well as various websites):

Eysenbach G

Improving the Quality of Web Surveys: The Checklist for Reporting Results of Internet E-Surveys (CHERRIES)

J Med Internet Res 2004;6(3):e34

doi: 10.2196/jmir.6.3.e34

I am not requiring that you use this format specifically, but it does look like a helpful checklist to use in order to report your survey data as completely as possible, without overlooking anything (e.g., you do not provide the timeframe/dates during which the survey in Study 1 was available).

Reviewer 2 suggests that while the paper is well-written (and I concur), you might consider reformatting it to help clarify the message. Again, I am not specifically requiring any particular format for dealing with both studies together in one manuscript, but I ask you to reconsider whether changing anything about the format will help deliver your message more clearly. I suspect that once you engage in ‘streamlining’ what gets presented in the text, the clarity will start to emerge on its own.

In summary, please revise this manuscript to reduce the content presented in the main text, as suggested by the reviewers and discussed above. I don’t feel it would be useful for me at this point to give you very specific instructions on what material should be moved, etc.! Rather, I think this revision should be more holistic, where you fully consider the reviewer comments and decide how to integrate the suggestions to produce a more readable and publishable manuscript. Of course, I also ask you to respond to the reviewers’ comments when your revision is submitted, and to indicate how their concerns have been addressed.

I hope you decide to revise and resubmit, as I do believe that these data are worthy of publication and address an area that needs much more study!

Best,

Carolyn

We look forward to receiving your revised manuscript.

Kind regards,

Carolyn J Walsh, PhD

Academic Editor

PLOS ONE

Journal Requirements:

"This study was funded, in part, by the Natural Sciences and Engineering Research Council of Canada and the British Columbia Society for the Prevention of Cruelty to Animals (#Industrial Research Chair 554745-19). We would like to extend our gratitude towards the members of the Animal Welfare Program at the University of British Columbia and the Companion Animal Welfare Lab for their feedback and encouragement"

"This study was funded, in part, by the Natural Sciences and Engineering Research Council of Canada and the British Columbia Society for the Prevention of Cruelty to Animals (#Industrial Research Chair 554745-19).

https://www.nserc-crsng.gc.ca/index_eng.asp
https://spca.bc.ca/

The authors Karen van Haaften and Amy Morris were employees of the BC SPCA at the time of the study. Both of the authors played a role in study design, data collection, and preparation of the manuscript"

"The authors Karen van Haaften and Amy Morris were employees of the BC SPCA at the time of the study"

We note that you received funding from a commercial source: BC SPCA

Reviewers' comments:

Reviewer's Responses to Questions

**Comments to the Author**

1. Is the manuscript technically sound, and do the data support the conclusions?

Reviewer #1: Partly

Reviewer #2: Yes

2. Has the statistical analysis been performed appropriately and rigorously? 

Reviewer #1: No

Reviewer #2: Yes

3. Have the authors made all data underlying the findings in their manuscript fully available?

Reviewer #1: Yes

Reviewer #2: Yes

4. Is the manuscript presented in an intelligible fashion and written in standard English?

Reviewer #1: No

Reviewer #2: Yes

5. Review Comments to the Author

Reviewer #1: Manuscript: PONE-D-21_25410

Investigation in owner-reported differences between dogs born in versus imported into Canada

General comments

- A timely study of current interest to a wider range of audiences. However, the paper itself does not seem to specifically set up to answer the aims of the study, and throughout the methods and results of study one, it feels more like it is a study that has been set up to generally identify a wide range of demographic factors that affect a wide range of dimensions of the dog – human owner relationship. There is much emphasis on other significant variables when I presume the key reason for including these if this study was genuinely designed to answer the aims set out would be to account for potential partial confounds and / or explanatory variables in the type of people that adopt overseas dogs or characteristics of these dogs that might act as alternative explanations for any significant difference found. i.e. you have run the multivariate analysis to find out whether, when accounting for these other variables, originating location of the dog adopted (overseas versus home grown) is significant. That does not come across at all in either the methods or the results of study one. Study two links more clearly to overseas dogs, but the quality of methodological reporting and claims made about replicating need revisiting.

- I cannot recommend this paper for acceptance without a major overhaul and rewrite. It is just far too much information, far too long, and far too unfocused at present. I think that the author has some interesting findings and the underpinning study has merit so it is worth persisting with this but it needs considerable revision.

- I also have major concerns about the sampling overlap between the two studies which I don’t think the author addresses through their reporting. If study 2 replicates and extends, why not just write up study two and accept that in study one you should have asked some key demographic information but failed to do so? Therefore it was necessary to repeat the study, and then treat study A as behind the scenes preliminary work (or a different paper with different aims – I am not entirely convinced this study was set up to specifically address the aims claimed over it). Or, did study B not replicate? The results suggests it might not have.

Abstract

- Clear abstract that contains all key components.

-

Introduction

- Line 68: “incomings” reads oddly. Perhaps consider: “imported”.

- Line 102 – 104. I find this very confusing having an aim embedded in the middle of the introduction. Aims belong at the end of an introduction, not early in the process when you are setting the scene/providing the background to the work. Don’t make someone have to read your full introduction to appraise your research study – they often don’t have time.

- Lines 105 – 18: Would be useful to a citation to support each of these measurement tools, ideally the citation that links back to the seminal paper for this.

- Overall, while interesting, I felt that this introduction was too long and felt a bit like an awkward blend of a literature review and an introduction. There is too much detail on each of the scales that the authors propose to use. I am estimating this introduction is about 1500 words long. I would estimate that cutting this down to around 750 – 1000 words would result in a much more tightly presented and succinct introduction.

- Line 180: I am not convinced at this stage that you can demonstrate this is a separate set of respondents.

Materials & methods

- Would be useful to see the methods (and results) reported in conjunction with CHERRIES reporting guidelines for online surveys. This would help with ensuring comprehensiveness of reporting.

- I find the division of material into the sub-headings and the subheadings themselves quite confusing. Some of this is probably down to lack of numbering of headings and sub-headings so that it is not clearly signposted what is a sub-sub-section of which sub-section and which is not. However, I also find the allocation of content to each section sometimes confusing. For example, study 1, methods starts with a collection of detailed information which is a mix of population, survey design, and ethical review. A more conventional restructure of the methods would make this much easier to follow.

- As a paid for panel was utilised, I assume that you know the number of individuals approached and therefore can calculate a response rate? What determined the number of people originally approached? Was this based on a sample size calculation? What company was used for the paid panel of online respondents? What was the wider demographic beyond age/education/gender? E.g. does the company only store details of dog owners, only store details of dog owners who do agility, etc or was it a much more disparate, not necessarily dog focused population that were sampled? I suspect the latter but this information needs to be provided.

- Using CHERRIES will give you more clues as to the level of additional detail I would like to see for the methods, but things like how many pages long was the survey, how many questions did it include, etc would be useful as part of the methods rather than just referring the reader to the appendices.

- Separating out the survey sections info from the scales used means that I am unable to visualise easily the survey design and how this fits together with the scales. Of course, I can jump across to the appendices to do this but it would be much kinder on the reader if the way that the methods are reported allows the reader to visualise the broad key structure from the methodological section. The appendices are to allow the reader to look for very specific additional information, but the methods should be sufficient for the reader to be able to visual the study design. From the way currently presented, I cannot do this.

- Statistical reporting – from what you have reported this seems reasonable however, the big issue here for me is that I have read the statistics section several times now and I cannot clearly see from this how you are answering your research question. Obviously you are, but it is not being communicated clearly in a way that lets the reader see this. I assume your most important variable of interest is: was the owned dog imported from overseas? Yes / no, given your research question/focus but that is not coming across in all the detail provided in your methods section. I was left wondering whether you had plotted this as an outcome / dependent or independent measure. Basically in all the detailed reporting of what you had done, the key point of it all seems to be missing. This obviously needs to be addressed.

- Study 2:

- Line 793 onwards: so what measures were taken to screen out those from the panel that had already completed the survey the first time? How many people made up the panel population that were eligible and, of these, what % were approached? i.e. was there considerable overlap in the sample between study 1 and 2?

- I am confused. This study seemed to be to replicate/extend study 1’s methods/results, but I see no information about the scales and these are not reported. My concerns more generally about methods reporting remain for this study too.

- Response rates, excluded responses, etc are normally reported at the start of the results.

Results

- Line 265: response rate? How many responses were not included in the survey? Why?

- Line 265: “51.7%”. Please don’t start a sentence with a numeral. This sentence should be restructured. E.g. “Approximately half (51.7%)”.

- This is a lot of tables to report demographic information, it might be worth merging into one table per study?

- I am confused by the results material given over to reporting the factor analysis process and outcomes. This was described as part of the statistical preparation so I assumed this was behind the scenes methods designed to condense the data ready for inferential analyses. However, it is now being treated like results in an off itself. I think it is appropriate to report how questions loaded, etc onto dimensions – but not as results. It is methods, or, possibly supplementary material. Here in the results section it just detracts from answering the central aims of your study – which should have been answered immediately following reporting your sample size/response rate, and demographic data.

- Some of the Cronbach alphas for the data are very low e.g. 0.61 (line 329) and 0.55 (line 319) with questionable or poor consistency. This does not seem to have been noted or measures taken to try to improve the CA (e.g. remove an item and re-run the analysis).

- Again, line 340 onwards. Some of this is methods and improved clarity in the methods section would remove the need to report methods within the results section. Stick to reporting the results, with clearer signposting as to what aim is being answered with which section of results reporting.

- One thing that is not clear is how the aims of your study are being answered. It is lost behind a scattergun reporting of anything that is significant. A classic example of this is found on page 28 where origin was significant yet you chose to report the other significant variables first (some of them), and bury origin in the middle.

- It is difficult to follow the tables e.g. table 11, to see what parameters are significant without additional information on the variable sub-categories. E.g. what does ‘Canadian’ denote? I have to try to pick it apart and make assumptions as I read through the other analyses but this is not ideal and make mean I am making unwarranted assumptions about how the data was handled.

- Line 363: when you report a p value it is either p = X, where X is the exact probability, or p < X, where the actual probability is less than X. You have written p = < 0.001. This needs addressing here and anywhere else that this has been done.

- Table 11: so what is the direction of any effect? I know, for example, that dog age and education level predicted C-BARQ, but how?

- Overall, to summarise study 1’s results section: this is just too much results and too little focus. It needs cutting back and restructuring to focus on clearly addressing the study’s aims. There are 27 pages of results for study 1 alone!

Discussion

- Reasonable but I have not drilled down on this because, in light of my above comments, I am recommending that this paper needs a major overhaul, and the discussion will need to be reformatted in conjunction with this.

Tables & figures

- Far too many, it is overwhelming with content and they are often not provided with enough detail to appreciate any direction of effect.

Reviewer #2: Strong article from authors on a very important topic. One question I did have reading through the long introduction, is whether the study could be split into two? Just a question, would like to know the reasoning behind two studies in a single journal article.

Results:

288: I do not know about the other reviewer, however, I would considering condensing the Cronbach’s alpha tables and section in the results to a summary and place the tables in appendix if possible. It will decrease the results section, decrease tables, but also allow readers to focus on the important information. To be fair, I went straight to ODR dimensions.

289-302 – I think that those two paragraphs give the reader enough information on the EFA factors and reliability, review this section to add addition intext data then remove tables would be my recommendation.

340 - 445 Is there any way to condense the number of tables in this section? Maybe summary all significant data in a single table and place rest of tables in appendix ? Allows the readers to get to focus on text, with a quick summary table.

446-610 I would review all tables and identify a way to condense all down

The paper is well written; however, all the data takes away from the story, from what you are trying to say. There is a LOT of data, and I enjoy the data however, most readers will be thrown.

Before I review any further, I would like authors to firstly revise results section, and secondly find a way to either split the two studies in part 1 and part 2, or condense the paper to focus on the most significant data.

Another format that could be appropriate is

Methods:

Study 1

Study 2

Results:

Study 1

Study 2:

Discussion:

General discussion

Study 1

Study 2

If you go this way it will flow a lot more.

Overall, the paper is well written and the statistical model used are excellent. Just needs to be more reader friendly.

6. PLOS authors have the option to publish the peer review history of their article (what does this mean?). If published, this will include your full peer review and any attached files.

Reviewer #1: No

Reviewer #2: No

---

## [Author Response · Author response to Decision Letter 0]

28 Dec 2021

Response to Reviewers 

Manuscript: PONE-D-21_25410

I would like to thank all reviewers for taking the time to read this manuscript and for their thoughtful suggestions. The suggestions that were made were extremely helpful in guiding the editing process and improving the overall quality of this report. Please find our comments to the suggestions below. Any edits that were to the manuscript can be found in the “Revised Manuscript with Track Changes” document. 

Reviewer #1: 

- A timely study of current interest to a wider range of audiences. However, the paper itself does not seem to specifically set up to answer the aims of the study, and throughout the methods and results of study one, it feels more like it is a study that has been set up to generally identify a wide range of demographic factors that affect a wide range of dimensions of the dog – human owner relationship. There is much emphasis on other significant variables when I presume the key reason for including these if this study was genuinely designed to answer the aims set out would be to account for potential partial confounds and / or explanatory variables in the type of people that adopt overseas dogs or characteristics of these dogs that might act as alternative explanations for any significant difference found. i.e. you have run the multivariate analysis to find out whether, when accounting for these other variables, originating location of the dog adopted (overseas versus home grown) is significant. That does not come across at all in either the methods or the results of study one. Study two links more clearly to overseas dogs, but the quality of methodological reporting and claims made about replicating need revisiting.

Thank you for the comment. We decided to include as much of our findings into the original manuscript as possible as there is currently limited research available in this field. However, we understand how this may have been a source of confusion and frustration for the readers. The manuscript has been edited to remove less pertinent information and instead, to only present information directly relevant to dog importation. 

- I cannot recommend this paper for acceptance without a major overhaul and rewrite. It is just far too much information, far too long, and far too unfocused at present. I think that the author has some interesting findings and the underpinning study has merit so it is worth persisting with this but it needs considerable revision.

Thank you. Significant efforts have been made to reduce the length of the manuscript as much as possible. Less relevant tables and figures has been removed or relocated, and wording has been revised. We hope this improves the readability and focus of the report.

- I also have major concerns about the sampling overlap between the two studies which I don’t think the author addresses through their reporting. If study 2 replicates and extends, why not just write up study two and accept that in study one you should have asked some key demographic information but failed to do so? Therefore it was necessary to repeat the study, and then treat study A as behind the scenes preliminary work (or a different paper with different aims – I am not entirely convinced this study was set up to specifically address the aims claimed over it). Or, did study B not replicate? The results suggests it might not have.

Thank you for the comment. Study 2 was not a replication of Study 1. Study 1 investigated a broader spectrum of questions related to dog importation. The aim of Study 2 was to confirm our results from Study 1 and provide additional information. 

Abstract

- Clear abstract that contains all key components.

Great, thank you!

Introduction

- Line 68: “incomings” reads oddly. Perhaps consider: “imported”.

Thank you for spotting the error. This section of the introduction was revised; thus, the suggestion is no longer applicable. Corresponding section can be found in line 65-69. 

- Line 102 – 104. I find this very confusing having an aim embedded in the middle of the introduction. Aims belong at the end of an introduction, not early in the process when you are setting the scene/providing the background to the work. Don’t make someone have to read your full introduction to appraise your research study – they often don’t have time.

This is a great point. We agree that this formatting may be a source of confusion and frustration. Changes were made to relocate the aim to the end of the introduction section (line 138-140).

- Lines 105 – 18: Would be useful to a citation to support each of these measurement tools, ideally the citation that links back to the seminal paper for this.

Thank you for this suggestion. Citations were added for each measurement tool (line 131-134).

- Overall, while interesting, I felt that this introduction was too long and felt a bit like an awkward blend of a literature review and an introduction. There is too much detail on each of the scales that the authors propose to use. I am estimating this introduction is about 1500 words long. I would estimate that cutting this down to around 750 – 1000 words would result in a much more tightly presented and succinct introduction.

Thank you for your comment. The introduction has been revised to significantly reduce the length (from ~1700 words to 956 words). 

- Line 180: I am not convinced at this stage that you can demonstrate this is a separate set of respondents.

Thank you for the suggestion. While we agree that we cannot show definitively that not a single participant took both surveys, however this is highly unlikely as the studies used different surveying platforms and were in different time periods. We clarified this point in the manuscript (line 1222-1225).

Materials & methods

- Would be useful to see the methods (and results) reported in conjunction with CHERRIES reporting guidelines for online surveys. This would help with ensuring comprehensiveness of reporting.

Thank you for sharing such a wonderful resource. The CHERRIES guideline was reviewed, and changes were made to the manuscript to follow the guideline to the extent possible. Since the complete unique site visitor and view rate statistics were not available, the participation rate was not reported in the revised manuscript. However, the recruitment and response rate section was revised to adhere to the CHERRIES guideline as much as possible. Response rates are mentioned at the start of the Results section (line 381-384, and 1269-1274). 

- I find the division of material into the sub-headings and the subheadings themselves quite confusing. Some of this is probably down to lack of numbering of headings and sub-headings so that it is not clearly signposted what is a sub-sub-section of which sub-section and which is not. However, I also find the allocation of content to each section sometimes confusing. For example, study 1, methods starts with a collection of detailed information which is a mix of population, survey design, and ethical review. A more conventional restructure of the methods would make this much easier to follow. 

Thank you for your suggestion. The structure of the Methods section was reworked to reduce confusion and improve the flow. The Methods sections now include four sub-headings for Study 1: Survey design, Recruitment and respondents, Statistical preparation, and Statistical analysis (line 150-377). Study 2 contains three sub-headings: Survey design, Recruitment and respondents, and Statistical analysis (line 1194-1240). 

- As a paid for panel was utilised, I assume that you know the number of individuals approached and therefore can calculate a response rate? What determined the number of people originally approached? Was this based on a sample size calculation? What company was used for the paid panel of online respondents? What was the wider demographic beyond age/education/gender? E.g. does the company only store details of dog owners, only store details of dog owners who do agility, etc or was it a much more disparate, not necessarily dog focused population that were sampled? I suspect the latter but this information needs to be provided.

Recruitment information was retrieved to the extent possible. Unfortunately, not all statistics regarding recruitment was available, since participant invitations were sent by the survey companies (Study 1: SurveyGizmo, now titled Alchemer, Study 2: Stratcom) line 272-280, 381-384, 1220-1227, and 1269-1274.

- Using CHERRIES will give you more clues as to the level of additional detail I would like to see for the methods, but things like how many pages long was the survey, how many questions did it include, etc would be useful as part of the methods rather than just referring the reader to the appendices. Separating out the survey sections info from the scales used means that I am unable to visualise easily the survey design and how this fits together with the scales. Of course, I can jump across to the appendices to do this but it would be much kinder on the reader if the way that the methods are reported allows the reader to visualise the broad key structure from the methodological section. The appendices are to allow the reader to look for very specific additional information, but the methods should be sufficient for the reader to be able to visual the study design. From the way currently presented, I cannot do this.

Thank you for your suggestion. However, we would like to respectfully disagree with your suggestion as the manuscript was already lengthy, we have included both surveys into the Appendices to reduce the length. 

- Statistical reporting – from what you have reported this seems reasonable however, the big issue here for me is that I have read the statistics section several times now and I cannot clearly see from this how you are answering your research question. Obviously you are, but it is not being communicated clearly in a way that lets the reader see this. I assume your most important variable of interest is: was the owned dog imported from overseas? Yes / no, given your research question/focus but that is not coming across in all the detail provided in your methods section. I was left wondering whether you had plotted this as an outcome / dependent or independent measure. Basically in all the detailed reporting of what you had done, the key point of it all seems to be missing. This obviously needs to be addressed.

Thank you for your comment. This issue may have been a result of over-reporting of the data. To minimize confusion, findings not directly related to dog importation (interactions between factors; Correlation matrix, interactions between factors and dog care questions; Multiple t-test, and interactions between dog care questions; Chi-squared test) were removed from the manuscript. 

Study 2:

- Line 793 onwards: so what measures were taken to screen out those from the panel that had already completed the survey the first time? How many people made up the panel population that were eligible and, of these, what % were approached? i.e. was there considerable overlap in the sample between study 1 and 2?

There were no specific procedures to minimize overlap between the two studies. While it is unlikely that there is considerable overlap of participants approached between Study 1 and Study 2 since two different surveying platforms were used, there is no way to confirm this theory. This point is also included in the methods section for Study 2 of the manuscript (line 1222-1225). 

- I am confused. This study seemed to be to replicate/extend study 1’s methods/results, but I see no information about the scales and these are not reported. My concerns more generally about methods reporting remain for this study too.

Thank you for pointing out your confusion. Reporting of the scales and measurements used in Study 1 and Study 2 were revamped to improve clarity (line 155-269, and 1201-1212). The confusion may have also been partly due to the over-reporting, which should no longer be an issue.

- Response rates, excluded responses, etc are normally reported at the start of the results.

Thank you for this input. Details of response rates were included at the start of Results section for Study 1 (line 381-384) and Study 2 (line 1269-1274).

Results

- Line 265: response rate? How many responses were not included in the survey? Why?

Thank you for this comment. Details of response rates were included at the start of Results section for Study 1 (line 381-384) and Study 2 (line 1269-1274).

- Line 265: “51.7%”. Please don’t start a sentence with a numeral. This sentence should be restructured. E.g. “Approximately half (51.7%)”.

Thank you for pointing out this error. Appropriate changes were made to the sentence (line 384).

- This is a lot of tables to report demographic information, it might be worth merging into one table per study?

Thank you for the suggestion. While we have taken this into consideration, we have decided to keep all tables for demographic information as they were relevant to dog importation. Instead, tables in the Results section that were less pertinent to dog importation were removed to reduce the length of the manuscript.

- I am confused by the results material given over to reporting the factor analysis process and outcomes. This was described as part of the statistical preparation so I assumed this was behind the scenes methods designed to condense the data ready for inferential analyses. However, it is now being treated like results in an off itself. I think it is appropriate to report how questions loaded, etc onto dimensions – but not as results. It is methods, or, possibly supplementary material. Here in the results section it just detracts from answering the central aims of your study – which should have been answered immediately following reporting your sample size/response rate, and demographic data.

Thank you for your idea. The factor analysis tables were removed from the manuscript file and relocated into the Supplementary information document. 

- Some of the Cronbach alphas for the data are very low e.g. 0.61 (line 329) and 0.55 (line 319) with questionable or poor consistency. This does not seem to have been noted or measures taken to try to improve the CA (e.g. remove an item and re-run the analysis).

Great point. Reasoning behind inclusion of “Expectaton” and “Gentle training” factors despite low internal consistency has been elaborated in the Methods section (line 318-340).

- Again, line 340 onwards. Some of this is methods and improved clarity in the methods section would remove the need to report methods within the results section. Stick to reporting the results, with clearer signposting as to what aim is being answered with which section of results reporting.

Thank you for your suggestion. Major changes were made to the results section to streamline the reporting included in the Results section. Specifically, the results section was reworked to minimize confusion by removing less pertinent information. Reporting was reduced to include only data related to dog import related (i.e., Correlation matrix, Multiple t-test, and Chi-squared results have been removed).

- One thing that is not clear is how the aims of your study are being answered. It is lost behind a scattergun reporting of anything that is significant. A classic example of this is found on page 28 where origin was significant yet you chose to report the other significant variables first (some of them), and bury origin in the middle.

Good point. I see the source of confusion. This should no longer be an issue, as findings not pertinent to dog importation has been removed from the manuscript. 

- It is difficult to follow the tables e.g. table 11, to see what parameters are significant without additional information on the variable sub-categories. E.g. what does ‘Canadian’ denote? I have to try to pick it apart and make assumptions as I read through the other analyses but this is not ideal and make mean I am making unwarranted assumptions about how the data was handled.

Thank you for highlighting this potential source of confusion. Tables included in the Results section for Study 1 has undergone major changes so this comment is no longer applicable. The newly designed tables should not cause the same confusion as the categories are labeled under their corresponding variable (Table 7-15; line 524-743).

- Line 363: when you report a p value it is either p = X, where X is the exact probability, or p < X, where the actual probability is less than X. You have written p = < 0.001. This needs addressing here and anywhere else that this has been done.

Great point. I was not aware of this error, and I fully agree with your comment. All incorrect reporting of P value (P=<) has been changed to either P=X or P<X.

- Table 11: so what is the direction of any effect? I know, for example, that dog age and education level predicted C-BARQ, but how?

Thank you for this suggestion. The directionality of the effect of owner and dog parameters on the factors should now be apparent as Mean and SD values are included in the newly designed tables (Table 7-15; line 524-743).

Reviewer #2

- Overall, to summarise study 1’s results section: this is just too much results and too little focus. It needs cutting back and restructuring to focus on clearly addressing the study’s aims. There are 27 pages of results for study 1 alone!

Thank you for your suggestion. Major changes have been made to the Results section to reduce the length and improve the focus of the study. Reporting was reduced to include only data related to dog import related (correlation matrix, multiple t-test, and chi-squared data was removed). 

Discussion

- Reasonable but I have not drilled down on this because, in light of my above comments, I am recommending that this paper needs a major overhaul, and the discussion will need to be reformatted in conjunction with this.

Thank you for your comment. The discussion section has been reformatted to align with the findings presented in the Results section. The discussion now only includes material directly related to dog importation (Effect of dog origin, Pet store as source, Source as puppy, Household as source, and Effect of owner and dog parameters; line 888-1168). 

Tables & figures

- Far too many, it is overwhelming with content and they are often not provided with enough detail to appreciate any direction of effect.

Reviewer #2: Strong article from authors on a very important topic. One question I did have reading through the long introduction, is whether the study could be split into two? Just a question, would like to know the reasoning behind two studies in a single journal article.

Thank you for this suggestion. The number of tables has been reduced to only include data directly relevant to dog importation. Additionally, the mean and standard deviation values have been added to the tables to allow for the interpretation of directionality of effect (Table 7-15; line 524-743).

Results:

288: I do not know about the other reviewer, however, I would considering condensing the Cronbach’s alpha tables and section in the results to a summary and place the tables in appendix if possible. It will decrease the results section, decrease tables, but also allow readers to focus on the important information. To be fair, I went straight to ODR dimensions.

What a wonderful idea – thank you. The EFA and Cronbach alpha tables have been relocated into the Supplementary information document to reduce the length of the Results section. 

289-302 – I think that those two paragraphs give the reader enough information on the EFA factors and reliability, review this section to add addition intext data then remove tables would be my recommendation.

Thank you for your comment. This component has been relocated to the Methods section to improve the structure of the manuscript (line 316-344). Additionally, EFA tables have been removed and relocated to the Supplementary information document to reduce manuscript length. 

340 - 445 Is there any way to condense the number of tables in this section? Maybe summary all significant data in a single table and place rest of tables in appendix ? Allows the readers to get to focus on text, with a quick summary table.

Thank you for your comment. Significant efforts have been made to reduce the length of the manuscript, but we believe that the results presented in this section are too crucial to investigate the effect of source on dog welfare to remove from the manuscript. However, the wording of this section has been revised to reduce length as much as possible. Additionally, less pertinent reports originally included in the Results section (Correlation matrix, Multiple t-test, and Chi-squared figures) have been removed to reduce the length.

446-610 I would review all tables and identify a way to condense all down

Thank you for the suggestion. Most of this section have been removed from the manuscript entirely as they were not directly relevant to dog importation. Reporting of on Dog care questions (Vet visit, Professional help, Internet help, and Experience help) are still available as they directly examine the effect of dog origin (line 755-884). 

The paper is well written; however, all the data takes away from the story, from what you are trying to say. There is a LOT of data, and I enjoy the data however, most readers will be thrown. Before I review any further, I would like authors to firstly revise results section, and secondly find a way to either split the two studies in part 1 and part 2, or condense the paper to focus on the most significant data.

Dear Reviewer #2, thank you for your review and kind comments. The manuscript has been edited to reduce length and to focus the reporting to dog importation. Tables and figures have been streamlined to only report data directly relevant to dog importation. Factor analysis and Cronbach alpha tables were removed from the manuscript and relocated to the Supplementary information document. Wording of the manuscript have also been revised to convey only most significant findings.

---

## [Editor Report · Decision Letter 1]

7 Feb 2022

PONE-D-21-25410R1Investigation into owner-reported differences between dogs born in versus imported into CanadaPLOS ONE

Dear Dr. von Rentzell,

Thank you for submitting your manuscript to PLOS ONE. After careful consideration, we feel that it has merit but does not fully meet PLOS ONE’s publication criteria as it currently stands. Therefore, we invite you to submit a revised version of the manuscript that addresses the points raised during the review process.

Dear Kai-Thank-you for your careful revision of your manuscript. I believe that you have addressed the major concerns of both Reviewers, and, based on that, I did not ask them to review the revised manuscript. Instead, I have created a list of revisions that I ask you to complete prior to a final decision on your paper.  As you will see below, many of the revisions are of a very minor nature, but there are two areas that require slightly more substantial consideration:1) clarification of the demographic and owner-dog relationship (ODR) factors in the main body of your paper, and 2) reorganization of the data presented for Study 2. Line-by-line details are pasted below. Please address each point and indicate on resubmission how they have been dealt with. I look forward to your revision.  Best,

Carolyn**********Academic Editor Revisions Required:  Abstract

Lines 42-43= not necessary to insert the statistics as long as supporting results are in the body of the paper

Introduction

Line 85- italicize both Genus names

Study 1

Methods

Survey Design- Please further explain the differences between your very similarly-named factors “Dog Source” and “Puppy Source”, as well as “Household as source”, which you don’t make clear until the Discussion. The reader needs this information well in advance of the Discussion.

Does it makes any sense to rename your factor “Puppy Source” to something less similar to “Dog source”- e.g., maybe “Puppy Background”? Not required if clarification is made.

Also, it isn’t clear to me whether a dog who was obtained in Canada an adult, but whose owner reported them to be “born to dog from international breeder (intentional litter)” would be considered a Canadian or non-Canadian dog. This may not have happened, I suppose! But this question occurred to me based on lines 524-525 in the Discussion. I think it just speaks to the need to further clarify how your demographic questions were structured. I know that you have placed a paragraph about the dog parameters in the S1 Supporting Information, but particularly for the factors that you find affect the ODR dimensions in different directions, the reader needs ‘in text’ access to the different definitions. As well, in the S1 Supporting Information, you use the term “international rescue” which is not referred to in the main text as a factor, so doesn’t help clarify the distinctions.

Line 134- what does “This component…” refer to? Do you mean “These dimensions…” or only specific dimensions? Please clarify

Lines 169+- to support this paragraph and give the reader more clarity on the structure of the dimensions you are evaluating, I’d ask you to consider creating a new ‘definitions’ Table, for the Methods section, that lists and defines all the 11 factors to be explicitly evaluated in your upcoming analysis. This table could include a summary of the source of questions for each ODR dimension and the interpretation you give the Dimension. I am not suggesting that the details from the factor loadings belong in this table- I think putting them in the Supplementary Info as you have done makes more sense. 

Line 193- remove typo in between sentences

Results

Lines 211+ - the word “majority” appears in several places (line 221, line 225) without an article in front of it- i.e., either “a” or “the”; please check throughout text

Table 1-  move to Appendix- relevant results are reported in text

Table 2 and 4- Can you merge these tables for clarity? I.e., have the comparisons of ‘dog source’ for ALL 803 dogs in the study side-by-side the same sources for Non-Canadian dogs, unless there is a compelling reason not to do so. There doesn’t seem to be a need for two separate tables when the same sources are being considered.

Table 5- and all tables- be specific re: the parameter “age”- specify “Owner age”, as you do for Dog age in these tables.

Line 272-277- wording is awkward for many of the sentences describing the factors… Consider this re-writing: “Owners with…..education received,** and **make owners compared to female owners **reported higher “Struggle” scores.**
**Owners of** dogs that were acquired from a pet store compared to **those **obtained…..as a puppy **also reported** higher “Struggle” scores.”

Lines 281-284- the results regarding “Burden” here are quite confusing. This is related to the above comments regarding source of dog/ puppy background, I believe. If you clarify the demographic questions earlier this should help- however, you may need to rewrite these results sentences for further clarity.

Line 307  - remove “…resulted in higher “Regret” scores.”

Line 335- remove second “…scored higher”.

Line 375- replace “than” with “compared to”

Line 377- add “-s” to end of “score”

Line 414+ - Figure 1-  (and other Figures)- P values should be placed in Figure caption OR indicated with symbols on the figures (e.g. * p<.05, **p<.01, etc.)

Discussion

Line 474-475- rewrite to “…of veterinarians agreed **that they were seeing** more behavioural…”

Line 519+ - Household as source-  As indicated earlier the information from lined 522-525 needs to be presented more clearly earlier.  Because of these subtle differences in the two variables, I am struggling to make sense of the comparisons in the next paragraph.

Line 524- replace “unidentical” with “nonidentical”

Study 2

Methods

Survey Design- refer to the S2 appendix in the text around line 652 or earlier

Line 655- suggest rewrite as follows: "The inclusion criteria **were** identical to** those of** Study 1: **participants were required to be a current resident of British Columbia, Canada, to own **at least one dog**, and to **have owned that dog for at least six months."

Line 658- replace “from” with “compared to”

Line 659- consider adding “…and were separated by X months.”

Results

Tables need to be streamlined and/or removed. See comments below.

Table 16- REMOVE -results are reported in text

Lines 688-690+- Why is the information regarding size of Canadian dogs and purebred dog breeds in the text, but then the data for non-Canadian dogs are presented in exquisite detail in Tables? This seems to be an imbalance. At the very least, the table info for the Canadian dogs could be placed in the Appendix/SI.

However, I believe a far better way to handle these differences could be one large table that makes Canadian vs. Non-Canadian dog comparisons on some (or all?) of the factors you wish to highlight. Currently, the presentation of data for non-Canadian dogs occurs in too much detail that is interesting as Supplementary Information, perhaps, but not always germane to your take-home message from these data.

Tables 17, 18- REMOVE/Move- data could be further presented in text, OR combined into single comparative table as suggested above, OR placed in SI

Table 19- Move to SI only

Tables 20 & 21- present as part of a single comparative table as suggested above OR in SI

Table 22- Move to SI only

Line 742-743: rewrite: “Canadian dogs were **comprised of **31.1% large dogs **compared to non-Canadian dogs, of which 20.7% were large dogs.**”

Line 745- Add “-s” to “parameter” in Table 23 title

Figure 4- Is it possible to indicate on the figure (eg., with bolding or otherwise) which of the factors has significant ORs?

Limitations

Line 853- change “was” to “were”

In this section, it may be worth considering the limitations of restricting the study to residents of BC only, as owners in other provinces may show different patterns of obtaining international dogs that are quite specific. E.g., in Newfoundland and Labrador, there is a local group that imports retired racing greyhounds from the US, and one that brings in ‘saluki’-type/middle Eastern sight hounds from Quatar. In these cases, dogs could have very different backgrounds compared to dogs that are rescued as ‘street dogs’ from Central America, for example, which could impact the ODRs you are studying.

References

I have not provided detail on each reference, but the formatting is inconsistent with respect to case of the journal article title, in particular- so please revise accordingly to align with the journal’s author instructions. These references will be further checked by the journal editing team prior to proof production.

We look forward to receiving your revised manuscript.

Kind regards,

Carolyn J Walsh, PhD

Academic Editor

PLOS ONE
---

## [Author Response · Author response to Decision Letter 1]

7 Mar 2022

Abstract:

Lines 42-43 - not necessary to insert the statistics as long as supporting results are in the body of the paper 

- Thank you for your suggestion. Statistics have been removed from the abstract (line 42)

Introduction:

Line 85 - italicize both Genus names

- Thank you for your suggestion. Suggested changes have been made (line 95).

Study 1

Methods:

Survey Design - Please further explain the differences between your very similarly-named factors “Dog Source” and “Puppy Source”, as well as “Household as source”, which you don’t make clear until the Discussion. The reader needs this information well in advance of the Discussion. Does it makes any sense to rename your factor “Puppy Source” to something less similar to “Dog source”- e.g., maybe “Puppy Background”? Not required if clarification is made.

- Thank you for your suggestion. Further information has been added under the methods section for “Dog source” and “Puppy source” (now re-named as puppy background). Household as source was not a separate measurement, but rather a category for dog source and puppy background. We hope the newly added information will clarify the ambiguity in the measurements collected (lines 143-157).

Also, it isn’t clear to me whether a dog who was obtained in Canada an adult, but whose owner reported them to be “born to dog from international breeder (intentional litter)” would be considered a Canadian or non-Canadian dog. This may not have happened, I suppose! But this question occurred to me based on lines 524-525 in the Discussion. I think it just speaks to the need to further clarify how your demographic questions were structured. I know that you have placed a paragraph about the dog parameters in the S1 Supporting Information, but particularly for the factors that you find affect the ODR dimensions in different directions, the reader needs ‘in text’ access to the different definitions. 

- Thank you for highlighting this confusion. There were 13 owners of “Canadian” dogs which were “born to a dog from an international breeder (intentional litter)”. These responses were included in the analysis as “Canadian” dogs, but I understand how this may not have been clear to the reader. Further information has been added to clarify this ambiguity (lines 265-272).

As well, in the S1 Supporting Information, you use the term “international rescue” which is not referred to in the main text as a factor, so doesn’t help clarify the distinctions.

- Thank you for your comment. The text describing “international rescue” was accidentally removed during previous revisions. Description of “international rescue” has been restored (lines 157-161).

Line 134 - what does “This component…” refer to? Do you mean “These dimensions…” or only specific dimensions? Please clarify

- Thank you for your comment. The referred text has been re-worded to help clarify the confusion (line 167).

Lines 169+ - to support this paragraph and give the reader more clarity on the structure of the dimensions you are evaluating, I’d ask you to consider creating a new ‘definitions’ Table, for the Methods section, that lists and defines all the 11 factors to be explicitly evaluated in your upcoming analysis. This table could include a summary of the source of questions for each ODR dimension and the interpretation you give the Dimension. I am not suggesting that the details from the factor loadings belong in this table- I think putting them in the Supplementary Info as you have done makes more sense. 

- Thank you for your suggestion. A new definitions table has been created in the methods section (line 206). The table defines each of the ODR dimensions and lists the extracted factor from each of the dimensions. 

Line 193 - remove typo in between sentences

- Thank you for spotting this error. The typo has been removed (line 235).

Results:

Lines 211+ - the word “majority” appears in several places (line 221, line 225) without an article in front of it- i.e., either “a” or “the”; please check throughout text

- Thank you for your comment. Texts that contain the word “majority” has been revised and corrected accordingly (lines 273, 274, 1012, 1018, and 1257).

Table 1 - move to Appendix- relevant results are reported in text

- Thank you for your suggestion. The table showing non-Canadian dog origin was moved to S1 supporting information (line 279).

Table 2 and 4 - Can you merge these tables for clarity? I.e., have the comparisons of ‘dog source’ for ALL 803 dogs in the study side-by-side the same sources for Non-Canadian dogs, unless there is a compelling reason not to do so. There doesn’t seem to be a need for two separate tables when the same sources are being considered.

- Thank you for your suggestion – what a great idea! Tables 2 and 4 have been merged (line 311). 

Table 5 - and all tables- be specific re: the parameter “age”- specify “Owner age”, as you do for Dog age in these tables.

- Thank you for your suggestion. All tables that contained “age” has been revised and corrected to instead mention “Owner age” (lines 405, 463, 527, 569, 634, and 677).

Line 272-277 - wording is awkward for many of the sentences describing the factors… Consider this re-writing: “Owners with…..education received, and male owners compared to female owners reported higher “Struggle” scores. Owners of dogs that were acquired from a pet store compared to those obtained…..as a puppy also reported higher “Struggle” scores.”

- Thank you for your comment. The ambiguity in the phrasing of these sentences were a result of an attempt to reduce the length of the manuscript in earlier revisions. These texts have been revised to achieve the right balance between clarity and conciseness (lines 430-660).

Lines 281-284 - the results regarding “Burden” here are quite confusing. This is related to the above comments regarding source of dog/ puppy background, I believe. If you clarify the demographic questions earlier this should help- however, you may need to rewrite these results sentences for further clarity.

- Thank you for pointing out this confusion in the text. Your confusion is, in fact a correct interpretation of the current findings. Dog source and puppy background collected similar, yet slightly different aspect of the dog’s history, and confusingly, the effect of these measurements on the owner-dog relationship was not in agreement. Further clarification was added in the reporting of this section to aid this issue (lines 438-460, 494-501, 562-566).

Line 307 - remove “…resulted in higher “Regret” scores.”

- Thank you for spotting this error. The error has been corrected (line 501). 

Line 335 - remove second “…scored higher”.

- Thank you for spotting this typo. Typo has been corrected (line 560).

Line 375 - replace “than” with “compared to”

- Thank you for your suggestion. Changes have been made to the text accordingly (line 655).

Line 377 - add “-s” to end of “score”

- Thank you for your comment. Suggested changes have been made (line 657).

Line 414+ - Figure 1- (and other Figures)- P values should be placed in Figure caption OR indicated with symbols on the figures (e.g. * p<.05, **p<.01, etc.)

- Thank you for your suggestion. P values have been placed in the Figure caption (lines 717, 732, and 745).

Discussion:

Line 474-475 - rewrite to “…of veterinarians agreed that they were seeing more behavioural…”

- Thank you for your suggestion. Changes were made as suggested (line 775).

Line 519+ - Household as source- As indicated earlier the information from lined 522-525 needs to be presented more clearly earlier. Because of these subtle differences in the two variables, I am struggling to make sense of the comparisons in the next paragraph.

- Thank you for pointing out your confusion. As you have suggested, further clarification on “Household as source” was added earlier in the manuscript (lines 146-160). 

Line 524 - replace “unidentical” with “nonidentical”

- Thank you for your suggestion. The wording has been changed to “different” (line 827).

Study 2

Methods:

Survey Design - refer to the S2 appendix in the text around line 652 or earlier

- Thank you for your suggestion. S2 appendix was referenced in the Survey design section (line 948)

Line 655 - suggest rewrite as follows: "The inclusion criteria were identical to those of Study 1: participants were required to be a current resident of British Columbia, Canada, to own at least one dog, and to have owned that dog for at least six months."

- Thank you for your suggestion. Changes have been made accordingly (line 968).

Line 658 - replace “from” with “compared to”

- Thank you for your suggestion. Changes have been made to the text accordingly (line 971).

Line 659 - consider adding “…and were separated by X months.”

- Thank you for your suggestion. However, we have decided not to make this addition to text as the manuscript has lengthened with the changes that has been made in the current revision. The main concerns that were addressed by previous reviewers were regarding the length and scope of the manuscript being too large (line 972).

Results:

Tables need to be streamlined and/or removed. See comments below.

Table 16 - REMOVE -results are reported in text

- Thank you for your suggestion. Table 16 has been removed (line 1022).

Lines 688-690+ - Why is the information regarding size of Canadian dogs and purebred dog breeds in the text, but then the data for non-Canadian dogs are presented in exquisite detail in Tables? This seems to be an imbalance. At the very least, the table info for the Canadian dogs could be placed in the Appendix/SI. However, I believe a far better way to handle these differences could be one large table that makes Canadian vs. Non-Canadian dog comparisons on some (or all?) of the factors you wish to highlight. Currently, the presentation of data for non-Canadian dogs occurs in too much detail that is interesting as Supplementary Information, perhaps, but not always germane to your take-home message from these data.

- Thank you for this suggestion. I see your point and agree that the tables provide interesting data but may not be as relevant to the main point with just non-Canadian data. I have taken your advice and created new tables that shows both Canadian and non-Canadian data (lines 1023-1071).

Tables 17, 18 - REMOVE/Move- data could be further presented in text, OR combined into single comparative table as suggested above, OR placed in SI

- Thank you for your suggestion. Both tables 17 and 18 have been modified to include both Canadian and non-Canadian data (lines 1023-1065).

Table 19 - Move to SI only

- Thank you for your suggestion. Table 19 have been moved to S2 Supporting Information (line 1070).

Tables 20 & 21- present as part of a single comparative table as suggested above OR in SI

- Thank you for your suggestion. Table 20 (now table 17) have been modified to include both Canadian and non-Canadian data (line 1071). However, suggested changes were not able to be implemented for table 21 (now table 18) since only owners of non-Canadian dogs were asked for their dog source method. As a result, dog source methods for Canadian dogs are unavailable in Study 2. Additionally, while table 21 only shows non-Canadian dog source methods, we believe the information presented in this table are pertinent to the main interest of this research. Therefore, we have decided to keep this table in the main body of the paper (line 1100). 

Table 22 - Move to SI only

- Thank you for your suggestion. Table 22 have been moved to S2 Supporting information (line 1112).

Line 742-743 - rewrite: “Canadian dogs were comprised of 31.1% large dogs compared to non-Canadian dogs, of which 20.7% were large dogs.”

- Thank you for your suggestion. Suggested changes have been made to the text (line 1126).

Line 745 - Add “-s” to “parameter” in Table 23 title

- Thank you for your suggestion. Suggested changes have been made (line 1128).

Figure 4 - Is it possible to indicate on the figure (eg., with bolding or otherwise) which of the factors has significant ORs?

- Thank you for your suggestion. Asterisks have been added next to the statistically significant independent variables to assist visibility to readers. Figure legend has also been changed accordingly (line 1150)

Limitations:

Line 853 - change “was” to “were”

- Thank you for your comment. Changes were made to the text accordingly (line 1245).

In this section, it may be worth considering the limitations of restricting the study to residents of BC only, as owners in other provinces may show different patterns of obtaining international dogs that are quite specific. E.g., in Newfoundland and Labrador, there is a local group that imports retired racing greyhounds from the US, and one that brings in ‘saluki’-type/middle Eastern sight hounds from Quatar. In these cases, dogs could have very different backgrounds compared to dogs that are rescued as ‘street dogs’ from Central America, for example, which could impact the ODRs you are studying.

- Thank you for your suggestion. The limitation section has been revised to include potential biases resulting from respondents being limited to BC dog owners (lines 1238-1250). 

References:

I have not provided detail on each reference, but the formatting is inconsistent with respect to case of the journal article title, in particular- so please revise accordingly to align with the journal’s author instructions. These references will be further checked by the journal editing team prior to proof production.

- Thank you for your comment. The reference section has been reviewed and changes have been made to fit the PLOS ONE referencing style. To our knowledge, the only difference in the referencing style of this current paper to that of PLOS ONE, is the phrasing “Available from: URL” is instead styled as “Available: URL”. This can be changed manually if necessary. However, the change has not been made in the current version of the manuscript since the reference manager (Mendeley) overrides manual changes when references get updated.

---

## [Editor Report · Decision Letter 2]

11 May 2022

Investigation into owner-reported differences between dogs born in versus imported into Canada

PONE-D-21-25410R2

Dear Dr. von Rentzell,

We’re pleased to inform you that your manuscript has been judged scientifically suitable for publication and will be formally accepted for publication once it meets all outstanding technical requirements.

Kind regards,

Carolyn J Walsh, PhD

Academic Editor

PLOS ONE

Additional Editor Comments (optional):

Dear Kai-

This is Carolyn Walsh, the Academic Editor for your manuscript. Thank-you for your careful attention to the required changes and the thorough response that you provided. While I have made the decision to accept, here are a few minor points that I invite you to consider changing prior having the paper go to proof.

Line 187/Table 1- there is no definition for either “harsh” or “gentle” training. These should be provided somewhere for the reader.

Line 249- replace “~” with a dash; also in Line 603

Lines 265-276- this section is difficult to read (e.g., sentence beginning in line 266) and also refers incorrectly to Table 5 in line 270 vs. Table 4, I believe.

Line 363- replace “P” in “Primary” with lower case “p”

Line 430- add “-s” to “parameter” in subheading

Line 485- use of “adoptive parents” seems inappropriate here, as the term was not used previously (owners has been the term used throughout)- maybe replace with “dog adopters”?

Lines 624-626- use past tense of verbs

Lines 714-718- these sentences seem out of place; would they be better placed around Line 702?

Line 867-  In your section on Limitations, I would recommend that you remind the reader that the statistical analyses were exploratory in nature, and you may have some spurious findings as a result of that. This is fine, as exploratory analyses are intended to stimulate further confirmatory research and provoke new hypotheses.

Line 891- ‘first nations’ should be “First Nations”

Line 927- I think you should replace “a surprisingly high number” with a more modest phrase... I am not sure that 16/115 (14%) would be a high number of puppies- although still surprising!

Best,

Carolyn
---

## [Editor Report · Acceptance letter]

25 May 2022

PONE-D-21-25410R2 

Investigation into owner-reported differences between dogs born in versus imported into Canada 

Dear Dr. von Rentzell:

I'm pleased to inform you that your manuscript has been deemed suitable for publication in PLOS ONE. Congratulations! Your manuscript is now with our production department. 

Kind regards, 

on behalf of

Dr. Carolyn J Walsh 

Academic Editor

PLOS ONE